# LLM Alignment as Retriever Optimization: An Information Retrieval Perspective

**Bowen Jin** [1 2]  **Jinsung Yoon** [2]  **Zhen Qin** [3]  **Ziqi Wang** [1]  **Wei Xiong** [1]  **Yu Meng** [4]  **Jiawei Han** [1]  **Sercan Ö. Arık** [2]

## Abstract

Large Language Models (LLMs) have revolutionized artificial intelligence with capabilities in reasoning, coding, and communication, driving innovation across industries. Their true potential depends on effective alignment to ensure correct, trustworthy and ethical behavior, addressing challenges like misinformation, hallucinations, bias and misuse. While existing Reinforcement Learning (RL)-based alignment methods are notoriously complex, direct optimization approaches offer a simpler alternative. In this work, we introduce a novel direct optimization approach for LLM alignment by drawing on established Information Retrieval (IR) principles. We present a systematic framework that bridges LLM alignment and IR methodologies, mapping LLM generation and reward models to IR's retriever-reranker paradigm. Building on this foundation, we propose **LLM Alignment as Retriever Preference Optimization** (LARPO), a new alignment method that enhances overall alignment quality. Extensive experiments validate LARPO's effectiveness with 38.9 % and 13.7 % averaged improvement on AlpacaEval2 and MixEval-Hard respectively. Our work opens new avenues for advancing LLM alignment by integrating IR foundations, offering a promising direction for future research.

## 1. Introduction

Large Language Models (LLMs) (Achiam et al., 2023; Team et al., 2024a) have demonstrated remarkable capacities in a wide range of fields including conversational modeling (Zhao et al., 2023a), reasoning (Wei et al., 2022) and code generation (Jiang et al., 2024). Unlocking the full poten-

tial of LLMs while ensuring their ethical, safe, and high-quality performance hinges on effective alignment (Wang et al., 2023). However, existing reinforcement learning-based LLM alignment methods (*e.g.*, PPO (Ouyang et al., 2022)) involve multi-stage training and are challenging to optimize. To this end, direct LLM preference optimization methods (*e.g.*, DPO (Rafailov et al., 2024)) are proposed to simplify the alignment process.

In this work, we further enhance direct LLM preference optimization, focusing on bringing Information Retrieval (IR) perspectives (Tay et al., 2022). Striking parallels exist between IR methodologies and LLM alignment techniques (Lin et al., 2022). For example, IR's retriever-reranker framework, which uses a retriever for broad semantic matching to generate a candidate set and a reranker for fine-grained refinement, offers a compelling analogy to the Best-of-N approach in LLM alignment (Dong et al., 2023; Sessa et al., 2024). In this analogy, the LLM acts as the retriever, while the reward model serves as the reranker. Furthermore, the common use of dual-encoder architectures in both LLM generation and IR retrievers, coupled with the reliance on cross-encoder architectures in reward models and IR rerankers, further underscores this synergy. Leveraging established IR techniques offers the potential to develop novel, easily implementable LLM alignment methods grounded in IR principles, leading to improved alignment quality.

Despite the promising connections between LLM alignment and IR, a systematic exploration of this synergy remains lacking. Specifically, three key gaps exist: (1) a clear mapping between LLM alignment mechanisms and core IR principles has not been established; (2) empirical evaluations of LLMs through an IR lens are scarce; and (3) proven IR techniques like retriever optimization, hard negative mining, and candidate list construction are underutilized for LLM alignment. This paper directly addresses these gaps by systematically bridging LLM alignment and IR methodologies. Our contributions are fourfold:

- We introduce a comprehensive framework that connects LLM alignment techniques with the established IR principles, providing a new perspective on LLM alignment.

- We demonstrate the significance of three key IR prin-

---

[1]University of Illinois at Urbana-Champaign [2]Google Cloud AI Research [3]Google DeepMind [4]University of Virginia. Correspondence to: Bowen Jin <bowenj4@illinois.edu>.

*Proceedings of the 42nd International Conference on Machine Learning*, Vancouver, Canada. PMLR 267, 2025. Copyright 2025 by the author(s).

ciples - retriever optimization objectives, hard negative mining, and candidate list construction - for improving LLM alignment.

- Building on these insights, we propose a novel alignment method, **LLM A**lignment as **R**etriever **P**reference **O**ptimization (LARPO), which demonstrably enhances alignment quality, with 38.9 % and 13.7 % relative averaged improvement on AlpacaEval2 and MixEval-Hard.

- We conduct further empirical studies to evaluate LLM performance using IR metrics, analyzing the impact of various post-training techniques.

In summary, this work establishes a crucial link between IR and LLM alignment, offering both novel insights and practical methods for advancing the field.

## 2. An Information Retrieval Perspective on LLMs

### 2.1. Primer on information retrieval

Information retrieval systems (Zhu et al., 2023) typically employ a two-stage process involving retrievers (Zhao et al., 2024) and rerankers (Lin et al., 2022). The retriever, often implemented as a bi-encoder (Figure 1), efficiently identifies a large set of $(K)$ potentially relevant passages, denoted as $D_{\text{retrieval}}$, from a corpora $C$ given a query $q$. This is achieved using a coarse-grained similarity function, $p_{\text{retrieval}}(d|q) = \text{Enc}_q^T(q) \cdot \text{Enc}_d(d)$, where $\text{Enc}_q$ and $\text{Enc}_d$ represent the query and passage encoders respectively:

$$D_{\text{retrieval}}(q) = \{d \in C \mid \max_{\text{top-}K} p_{\text{retrieval}}(d|q)\}. \quad (1)$$

However, due to the scale of the corpus, retrievers might not accurately capture fine-grained query-passage similarity with the simple dot production interaction function. Therefore, rerankers, typically implemented with cross-encoder (Figure 1), are employed to refine the ranking of the retrieved passages $D_{\text{retrieval}}$. The reranker produces a smaller set $(k)$ of top-ranked passages, $D_{\text{rank}}$, using a fine-grained similarity function, $r_{\text{rank}}(q, d) = w \cdot \text{Enc}(q, d)$, where $w$ is a learnable linear layer. Here, reranker adopts cross-encoder with both query/passage as inputs and encoded together while retriever adopts dual encoder for separate query/passage encoding.

$$D_{\text{rank}}(q) = \{d \in D_{\text{retrieval}}(q) \mid \max_{\text{top-}k} r_{\text{rank}}(q, d)\}. \quad (2)$$

The resulting ranked passages are ordered such that $D_{\text{rank}}(q) = \{d_1, d_2, \ldots, d_k\}$ where $r_{\text{rank}}(q, d_1) \geqslant r_{\text{rank}}(q, d_2) \geqslant \cdots \geqslant r_{\text{rank}}(q, d_k)$.

### 2.2. LLMs as retrievers. Reward models as rerankers

During inference, an LLM generates a response $y$ given an input prompt $x$ by modeling the probability distribution $p_{\text{LLM}}(y|x)$. Assuming a fixed maximum sequence length $L$ and a vocabulary space $V$ (Li et al., 2024), the set of all possible responses can be defined as $Y = \{y : y(1)y(2)...y(L)|y(i) \in V\} \subseteq V^L$.

We can conceptualize this process through an IR lens (Tay et al., 2022). The prompt $x$ can be viewed as analogous to a query $q$, the set of all possible responses $Y$ can be treated as the corpus $C$, and the generated response $y$ can be considered as the retrieved passage $d$. Thus, given a prompt $x$, the LLM effectively acts as a retriever, searching for the most probable responses $D_{\text{LLM}}(x)$ from response space $Y$:

$$D_{\text{LLM}}(x) = \{y \in Y \mid \max_{\text{top-}K} p_{\text{LLM}}(d|x)\}. \quad (3)$$

where $p_{\text{LLM}}(y|x)$ is analogous to $p_{\text{retrieval}}(d|q)$ in IR.

This analogy is further supported by the LLMs' architecture. As illustrated in Figure 1, the generative modeling with LLMs can be interpreted as the matching process of a bi-encoder model. The prompt is encoded into a vector representation by LLM, while response tokens are represented as token embedding vectors. For each token position decoding, prompt embedding (obtained often from the hidden state of the last layer of the LLM) and vocabulary token embeddings are compared with a dot product, to determine the likelihood of a selected token for the response.

Furthermore, reward models $r_{\text{rm}}(x, y)$ (Lambert et al., 2024), which take both the prompt and response as input, function similarly to cross-encoders (*i.e.*, rerankers $r_{\text{rank}}(q, d)$ (Zhuang et al., 2023)) in IR. To enhance LLM performance, various inference-time strategies have been developed, including Best-of-N sampling (Stiennon et al., 2020) and majority voting (Wang et al., 2022). These can be interpreted as different configurations of retrievers and rerankers, as summarized in Appendix Table 5.

### 2.3. LLM tuning as retriever optimization

**Supervised fine-tuning as direct retriever optimization.** Retriever training, aiming for accurate retrieval, often employs contrastive learning with the InfoNCE loss (Oord et al., 2018) to maximize $P(d_{\text{gold}}|q)$ of retrieving the ground truth passage $d_{\text{gold}}$ given a query $q$. This can be expressed as:

$$\max \log P(d_{\text{gold}}|q) = \max \log \frac{\text{Enc}_d(d_{\text{gold}}) \cdot \text{Enc}_q(q)}{\sum_{j=1}^{|C|} \text{Enc}_d(d_j) \cdot \text{Enc}_q(q)}.$$

In the context of LLM alignment, supervised fine-tuning (SFT) aims to quickly adapt the model to a target task using prompt-response pairs $(x, y_{\text{gold}})$. SFT maximizes the

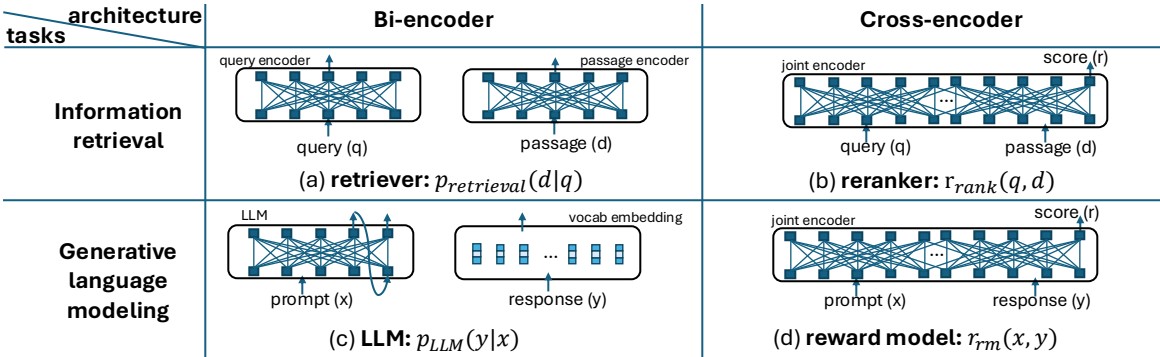

*Figure 1.* Architecture connection between retriever/LLM (bi-encoder) and reranker/reward model (cross-encoder). Bi-encoder models process each query/prompt and passage/response separately and often calculate their alignment score via a dot product operator, while cross-encoder models take both query/prompt and passage/response as input and score them directly. Bi-encoder models can be more efficient (*i.e.*, large-scale text matching) but the interaction between the two information unit is only captured by a dot production operation where their effectiveness can be constrained. Cross-encoder models can be more effective (*i.e.*, deeper interaction calculation with transformer architecture (Vaswani, 2017)) but less efficient. Although LLM involves auto-regressive token matching, which is different from retriever, some insights from IR can be borrowed to enhance LLM alignment as shown in the following sections.

conditional probability $P(y_{\text{gold}}|x)$ as:

$$\max \log P(y_{\text{gold}}|x) = \max \log \prod_{i}^{|y_{\text{gold}}|} P(y_{\text{gold}}(i)|z_i)$$

$$= \max \sum_{i}^{|y_{\text{gold}}|} \log \frac{\text{Emb}(y_{\text{gold}}(i)) \cdot \text{LLM}(z_i)}{\sum_{j=1}^{|V|} \text{Emb}(v_j) \cdot \text{LLM}(z_i)},$$

where $y(i)$ is the $i$-th token of $y$, $z_i = [x, y_{\text{gold}}(1 : i - 1)]$ represent the concatenation of the prompt $x$ and the preceding tokens of $y_{\text{gold}}$, $\text{LLM}(\cdot)$ produces a contextualized representation, and $\text{Emb}(\cdot)$ is the token embedding function. We assume vocab embeddings and LLM hidden states share the same dimension, as in most LLMs.

Consequently, the SFT objective can be interpreted as a composite of multiple retrieval optimization objectives. In this analogy, $\text{LLM}(\cdot)$ acts as the query encoder and $\text{Emb}(\cdot)$ serves as the passage (or, in this case, token) encoder.

**Preference optimization as reranker-retriever distillation.** In retriever training, optimizing solely based on query/ground-truth document pairs can be suboptimal, particularly when using in-batch negatives for efficiency. Performance can be enhanced by distilling knowledge from a more powerful reranker to retriever (Qu et al., 2020; Zeng et al., 2022). This distillation process can be represented as $f_{\text{rerank}}(\cdot) \xrightarrow{c} \text{data} \xrightarrow{g(\cdot)} f_{\text{retrieval}}(\cdot)$, where new data, generated by the reranker $f_{\text{rerank}}(\cdot)$ based on a rule $c$, is used to optimize the retriever $f_{\text{retrieval}}(\cdot)$ with an objective $g(\cdot)$.

Similarly, in LLM alignment, a preference alignment phase often follows supervised fine-tuning (SFT) to further enhance the model using an external reward model to absorb preferential supervision effectively. Methods like PPO

(Schulman et al., 2017) and iterative DPO (Guo et al., 2024) exemplify this approach. Here, the LLM (considered acting as the retriever) generates responses that are then scored by the reward model (considered acting as the reranker). These scores are used to create new training data, effectively performing distillation from the reward model into the LLM: $f_{\text{reward-model}}(\cdot) \xrightarrow{c} \text{data} \xrightarrow{g(\cdot)} f_{\text{LLM}}(\cdot)$. Thus, preference optimization can be viewed as a form of reranker-to-retriever distillation, analogous to the process used in traditional IR.

We conduct empirical studies to understand SFT and preference optimization from IR perspective in Appendix B and have further discussion in Appendices C and D.

### 2.4. Empirical insights into LLMs as IR models

**Evaluating LLMs as retrievers.** A common metric for evaluating retrievers is Recall@$N$, which assesses whether the top-$N$ retrieved passages include any relevant passages for a given query. In the context of LLMs, this translates to evaluating whether the top-$N$ generated responses contain a suitable response to the prompt, analogous to Pass@$N$ (Chen et al., 2021).

To draw the empirical connection between LLM and retrievers, we conduct an experiment on the GSM8K dataset (Cobbe et al., 2021) using Mathstral-7b-it (Mistral AI, 2025) and an experiment on the NQ dataset (Kwiatkowski et al., 2019) using e5 retriever. Figure 2 illustrates that increasing N can contribute to improved performance for both retriever and LLM. Detailed analysis can be found in Appendix E.

Greedy decoding, equivalent to $N = 1$, is a prevalent LLM inference strategy. However, as shown in Figure 2(b), allowing multiple attempts ($N > 1$) can substantially improve the chance of producing a correct answer, suggesting that per-

*Table 1.* LLM alignment objectives of LARPO. In the table, $\gamma(y \mid x) = \beta \log \frac{\pi_\theta(y|x)}{\pi_{\text{ref}}(y|x)}$. All the proofs can be found in Appendix F.

| Method | Assumption of $r(x,y)$ | Objective |
|---|---|---|
| DPO | $\Pr(y_w \geq y_l) = \sigma(r(x,y_w) - r(x,y_l))$ | $\mathcal{L}_{\text{pair}} = -\mathbb{E}\left[\log\sigma\left(\beta\log\frac{\pi_\theta(y_w|x)}{\pi_{\text{ref}}(y_w|x)} - \beta\log\frac{\pi_\theta(y_l|x)}{\pi_{\text{ref}}(y_l|x)}\right)\right]$ |
| LARPO (Contrastive) | $\Pr(y_w \geq y_l^{(1)},...,y_w \geq y_l^{(m)}) = \text{softmax}(r(x,y_w))$ | $\mathcal{L}_{\text{con}} = -\mathbb{E}\left[\log\frac{\exp\left(\gamma(y_w|x)\right)}{\exp\left(\gamma(y_w|x)\right) + \sum_{i=1}^m \exp\left(\gamma(y_l^{(i)}|x)\right)}\right]$ |
| LARPO (LambdaRank) | $\Pr(y_1 \geq ... \geq y_m) = \prod_{1<i<j<m}\sigma(r(x,y_i) - r(x,y_j))$ | $\mathcal{L}_{\text{lamb}} = -\mathbb{E}\left[\sum_{1<i<j<m}\log\sigma\left(\gamma(y_i \mid x) - \gamma(y_j \mid x)\right)\right]$ |
| LARPO (ListMLE) | $\Pr(y_1 \geq ... \geq y_m) = \prod_{i=1}^m \text{softmax}_i^m(r(x,y_i))$ | $\mathcal{L}_{\text{lmle}} = -\mathbb{E}\left[\sum_{i=1}^m \log\frac{\exp\left(\gamma(y_i|x)\right)}{\exp\left(\gamma(y_i|x)\right) + \sum_{j=i+1}^m \exp\left(\gamma(y_j|x)\right)}\right]$ |

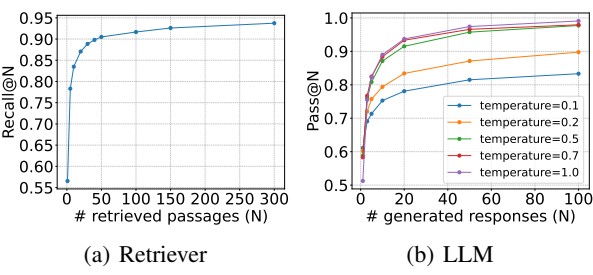

(a) Retriever    (b) LLM

*Figure 2.* Analogy between evaluating retriever with Recall@N and LLM with Pass@N. As the number (N) of retrieved passages/generated responses increases, the retriever and LLM have a similar increasing trend. This highlights the importance of inference time scaling (*e.g.*, Best-of-N) for LLM similar to retriever-reranker scaling in IR. Retriever: e5; LLM: Mathstral-7b-it.

formance under $N = 1$ may underestimate the model's full potential. This highlights the importance of inference-time scaling techniques like Best-of-N (Stiennon et al., 2020) in LLM similar to retriever-reranker scaling (Zhuang et al., 2023) in IR. More results and analyses can be found in Appendix E.

# 3. Iterative LLM alignment as retriever optimization

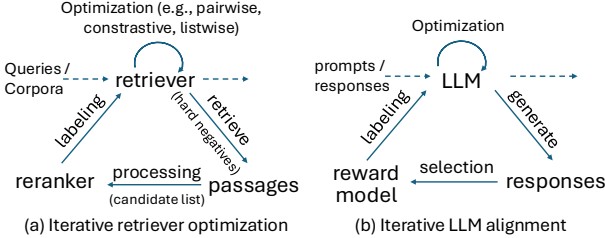

(a) Iterative retriever optimization    (b) Iterative LLM alignment

*Figure 3.* The connection between iterative LLM alignment (Xiong et al., 2024) and iterative retriever optimization (Xiong et al., 2020)

Iterative learning is a common technique in retriever optimization (Xiong et al., 2020), where results from the newly-trained model are used to generate new training data, as illustrated in Figure 3(a). Similarly, for LLM alignment, iterative preference optimization has been shown to enhance

performance (Xiong et al., 2024; Xu et al., 2024b; Guo et al., 2024) (Figure 3(b)). Drawing inspirations from retriever optimization, we re-examine iterative LLM preference optimization, focusing on three key aspects: (1) the optimization objective; (2) the use of hard negatives; and (3) the candidate list construction. Based on these aspects, we propose a new LLM alignment with an IR perspective, LARPO.

## 3.1. Retriever optimization objective

Typical objectives for retriever optimization include pairwise, contrastive and listwise objectives (Zhao et al., 2024). In this section, we discuss preference optimization variants (Wang et al., 2023) corresponding to different retriever optimization objectives. The optimization objective for preference optimization is given as:

$$\max_{\pi_{\text{LLM}}} \mathbb{E}_{x,y\sim\pi_{\text{LLM}}(\cdot|x)}[r(x,y)] - \beta\text{KL}(\pi_{\text{LLM}}(\cdot|x)||\pi_{\text{ref}}(\cdot|x)).$$

As discussed in Rafailov et al. (2024), the equation above has the optimal solution as:

$$r(x,y) = \beta\log\frac{\pi_{\text{LLM}}(y|x)}{\pi_{\text{ref}}(y|x)} + \beta\log Z, \tag{4}$$

where $Z = \sum_{y'}\pi_{\text{ref}}(y'|x)\exp(\frac{1}{\beta}r(x,y'))$ is the normalization constant and $r(\cdot)$ is the reward model which can also be seen as a reranker. According to different assumption for $r(x,y)$ from IR, we can obtain different training objectives as shown in Table 1, with proofs in Appendix F.

**Pairwise ranking.** Under the pairwise (Bradley-Terry) assumption $\Pr(y_w \geq y_l) = \sigma(r(x,y_w) - r(x,y_l))$, the policy objective becomes DPO (Rafailov et al., 2024) $\mathcal{L}_{\text{pair}}$.

**Contrastive ranking.** Another widely used objective for ranking is contrastive learning (Oord et al., 2018):

$$\Pr(y_w \geq y_l^{(1)},...,y_w \geq y_l^{(m)}) = \text{softmax}(r(x,y_w))$$
$$= \frac{\exp(r(x,y_w))}{\exp(r(x,y_w)) + \sum_{i=1}^m \exp(r(x,y_l^{(i)}))}. \tag{5}$$

It handles multiple negatives in a single step, allowing the model to learn more robust representations for retrieval

and ranking. It is widely used for dense retriever training (Karpukhin et al., 2020). Under this ranking assumption, the policy objective becomes $\mathcal{L}_{\text{con}}$ as shown in Table 1.

**LambdaRank.** In addition to pairwise and contrastive learning, list-wise ranking is widely adopted to sufficiently utilize the comprehensive information in candidate list. Inspired by LambdaRank (Burges, 2010; Zeng et al., 2022):

$$\Pr(y_1 \geq ... \geq y_m) = \prod_{1 < i < j < m} \sigma(r(x, y_i) - r(x, y_j)),$$
(6)

the policy optimization objective becomes $\mathcal{L}_{\text{lamb}}$ (Table 1).

**ListMLE.** Another list-wise ranking assumption is the ListMLE assumption (Xia et al., 2008), which provides theoretical grounding and global optimization perspective:

$$\Pr(y_1 \geq ... \geq y_m) = \prod_{i=1}^{m} \text{softmax}_i^m(r(x, y_i))$$
$$= \prod_{i=1}^{m} \frac{\exp(r(x, y_i))}{\exp(r(x, y_i)) + \sum_{j=i+1}^{m} \exp(r(x, y_j))}$$
(7)

In this case, the objective becomes $\mathcal{L}_{\text{lmle}}$ shown in Table 1.

### 3.2. Hard negatives

Hard negatives are crucial for effective retriever training (Zhan et al., 2021; Qu et al., 2020), as learning to distinguish harder negatives potentially lead to more powerful retrievers (Xiong et al., 2020). In LLM alignment, negatives correspond to unpreferred responses ($y_l$) for a given prompt ($x$). In iterative on-policy training, various types of negatives can be identified, ordered by increasing difficulty: (1) **Easiest**: A random, unrelated response to $x$; (2) **Easy**: A response to a related but different prompt ($x'$); (3) **Hard**: An incorrect response to $x$ generated with a high temperature; (4) **Hardest**: An incorrect response to $x$ generated with a low temperature.

Note that, assuming a well-initialized policy LLM, as indicated by Figure 2(b) ($N = 1$), low temperatures tend to produce harder negatives, yielding the above ranking. To be specific, lower temperatures yield more similar generated responses, increasing overlap between positive and negative samples. This effectively makes the negatives harder. According to Zhan et al. (2021), hardest negatives could be most important to LLM alignment.

### 3.3. Candidate list

In iterative retriever optimization, construction of the candidate list $[d_1, ..., d_m]$, which is used by the reranker to generate data for the next iteration, is crucial. Prior research (Zeng et al., 2022) has identified factors such as list

---

**Algorithm 1** LARPO: LLM alignment as iterative retriever preference optimization.

**Require:** Number of iterations $T$, number of new data per annotation phase $M$, number of generated responses for each prompt $k$, temperature for each iteration $\{t_i\}_{i=0}^{T}$, prompt dataset $\mathcal{D}_{\mathcal{X}} = \{x_i\}_{i=1}^{N}$, policy LLM $\pi_{\theta_0}$, reward model $r$, learning rate $\gamma$, a ranking-based objective function $\mathcal{L}_{\text{rank}}$.

**Ensure:** Aligned LLM $\pi_{\theta_T}$.

1: **for** $s := 0$ to $T$ **do**
2:     Update behavior LLM: $\pi_\beta \leftarrow \pi_{\theta_s}$
3:     Preference dataset $\mathcal{D}_s = \{\}$
4:     **for** $i := 1$ to $M$ **do**
5:         Sample prompt $x \sim \mathcal{D}_{\mathcal{X}}$
6:         // candidate list construction
7:         Sample $y_1, ..., y_k \sim \pi_\beta(\cdot|x)_{t_s}$
8:         // hard negatives
9:         Rank $\{y_i\}$ with $r$: $Y_x = \{y_j^{(r)}\}$, where $(r(y_a^{(r)}) > r(y_b^{(r)}))$, $a < b$
10:       $\mathcal{D}_s \leftarrow \mathcal{D}_s \cup \{(x, Y_x)\}$
11:     **end for**
12:     // candidate list construction
13:     $\mathcal{D} \leftarrow \text{Merge}_{k=0}^{s} \mathcal{D}_k$
14:     **while** $\mathcal{D} \neq \varnothing$ **do**
15:         Sample a batch $(x, Y_x)$ from $\mathcal{D}$
16:         Update $\mathcal{D} \leftarrow \mathcal{D} \backslash \{(x, Y_x)\}$
17:         // retriever optimization objective
18:         $\theta_s \leftarrow \theta_s - \gamma \cdot \nabla_\theta \mathcal{L}_{\text{rank}}(x, Y_x, \pi_\theta; \pi_\beta)$
19:     **end while**
20:     $\theta_{s+1} \leftarrow \theta_s$
21: **end for**

---

size and candidate selection as being particularly important. Similarly, in iterative preference optimization, construction of the candidate response list $Y = [y_1, ..., y_m]$ is critical. We identify two key factors influencing the quality of $Y$: inclusiveness and memorization.

(1) **Inclusiveness** (Qu et al., 2020) refers to the **size** of the response list $Y$. A larger $Y$ potentially encompasses more information.

(2) **Memorization** (Zeng et al., 2022) refers to whether previously generated responses $Y'$ are included in the current list $Y$ to preserve past results.

Given their importance in IR (Qu et al., 2020; Zeng et al., 2022), the impact of these factors on LLM alignment, however, remains largely under-explored.

## 4. The Proposed Solution: LARPO

Motivated by iterative retriever optimization pipeline as

*Table 2.* Evaluations on AlpacaEval 2 and MixEval. LC WR and WR denote length-controlled win rate and win rate respectively. Offline baseline performances on AlpacaEval 2 are from Meng et al. (2024b). We use LLM-blender (Jiang et al., 2023b) as the reward model for a fair comparison with the baselines and also report the result with a stronger reward model FsfairX (Dong et al., 2024)

| Model | Mistral-Base (7B) | | | | Mistral-Instruct (7B) | | | |
|---|---|---|---|---|---|---|---|---|
| | Alpaca Eval 2 | | MixEval | MixEval-Hard | Alpaca Eval 2 | | MixEval | MixEval-Hard |
| | LC WR | WR | Score | Score | LC WR | WR | Score | Score |
| SFT | 8.4 | 6.2 | 0.602 | 0.279 | 17.1 | 14.7 | 0.707 | 0.361 |
| Reward model: LLM-Blender (Jiang et al., 2023b) | | | | | | | | |
| RRHF | 11.6 | 10.2 | 0.600 | 0.312 | 25.3 | 24.8 | 0.700 | 0.380 |
| SLiC-HF | 10.9 | 8.9 | 0.679 | 0.334 | 24.1 | 24.6 | 0.700 | 0.381 |
| DPO | 15.1 | 12.5 | 0.686 | 0.341 | 26.8 | 24.9 | 0.702 | 0.355 |
| IPO | 11.8 | 9.4 | 0.673 | 0.326 | 20.3 | 20.3 | 0.695 | 0.376 |
| CPO | 9.8 | 8.9 | 0.632 | 0.307 | 23.8 | 28.8 | 0.699 | 0.405 |
| KTO | 13.1 | 9.1 | **0.704** | 0.351 | 24.5 | 23.6 | 0.692 | 0.358 |
| RDPO | 17.4 | 12.8 | 0.693 | 0.355 | 27.3 | 24.5 | 0.695 | 0.364 |
| SimPO | 21.5 | 20.8 | 0.672 | 0.347 | 32.1 | 34.8 | 0.702 | 0.363 |
| Iterative DPO | 18.9 | 16.7 | 0.660 | 0.341 | 20.4 | 24.8 | 0.719 | 0.389 |
| LᴀRPO (Contrastive) | 31.6 | 30.8 | 0.703 | 0.409 | 32.7 | 38.6 | 0.718 | **0.418** |
| LᴀRPO (LambdaRank) | **34.9** | **37.2** | 0.695 | **0.452** | **32.9** | **38.9** | **0.720** | 0.417 |
| LᴀRPO (ListMLE) | 31.1 | 32.1 | 0.669 | 0.390 | 29.7 | 36.2 | 0.709 | 0.397 |
| Reward model: FsfairX (Dong et al., 2024) | | | | | | | | |
| LᴀRPO (Contrastive) | **41.5** | **42.9** | 0.718 | 0.417 | **43.0** | **53.8** | 0.718 | 0.425 |
| LᴀRPO (LambdaRank) | 35.8 | 34.1 | 0.717 | 0.431 | 41.9 | 48.1 | **0.740** | **0.440** |
| LᴀRPO (ListMLE) | 36.6 | 37.8 | **0.730** | **0.423** | 39.6 | 48.1 | 0.717 | 0.397 |

shown in Figure 3(a) and the three key points in IR, we introduce LᴀRPO, a novel approach to LLM alignment formulated as iterative retriever preference optimization. The algorithmic details are provided in Algorithm 1. Specifically, our experimental setup explores the following key aspects: (1) **Optimization objective**: We evaluate three distinct loss functions as the ranking objective ($\mathcal{L}_{rank}$): $\mathcal{L}_{con}$, $\mathcal{L}_{lamb}$, and $\mathcal{L}_{lmle}$. (2) **Hard negatives**: For a given prompt, hard negative samples are constructed by selecting less preferred responses generated with an appropriate temperature through parameter search. More details of the temperature are available in Appendix H.1. (3) **Candidate list**: In each iteration, we generate multiple (10) candidate responses considering inclusiveness. In terms of memorization, the candidate pool for subsequent iterations includes all previously generated responses.

## 5. Main Results

**Baselines.** We evaluate the performance of LᴀRPO against a range of established preference optimization methods, encompassing both offline and online approaches. Our offline comparison set includes RRHF (Yuan et al., 2023), SLiC-HF (Zhao et al., 2023b), DPO (Guo et al., 2024), IPO (Azar et al., 2024), CPO (Xu et al., 2024a), KTO (Ethayarajh et al., 2024), RDPO (Park et al., 2024) and SimPO (Meng et al., 2024b). For online methods, we compare with iterative DPO (Xiong et al., 2024). The baseline checkpoints

are from Meng et al. (2024b). Further details regarding these baselines and our experimental setup are provided in Appendix G. Both baselines and LᴀRPO are trained on Ultrafeedback dataset (Cui et al., 2024) for fair comparison.

**Datasets.** We conduct evaluation on two widely used benchmarks AlpacaEval2 (Dubois et al., 2024) and MixEval (Ni et al., 2024). These benchmarks are designed to assess the conversational capabilities of models across a diverse range of queries. AlpacaEval2 comprises 805 questions sourced from five datasets, while MixEval includes 4000 general and 1000 hard questions. Evaluation follows the established protocols for each benchmark. For AlpacaEval 2, we report both the raw win rate (WR) and the length-controlled win rate (LC). These benchmarks collectively provide a comprehensive assessment of the models' instruction-following and problem-solving capabilities.

**Results.** The baseline performances on AlpacaEval 2 are directly from Meng et al. (2024b), while the performances on MixEval is evaluated by ourselves with the opensourced checkpoints. We adopt the same LLM-Blender (Jiang et al., 2023b) reward model for a fair comparison with the baselines and also explore stronger reward model: FsfairX (Dong et al., 2024). The results, presented in Table 2, show that LᴀRPO consistently outperforms the competitive baseline methods on both datasets, with 38.9 % and 13.7 % averaged relative improvements, on AlpacaEval2 and MixEval-

Hard respectively, with the same reward model as the baselines. With a stronger reward model, we can further improve LARPO by 25.8 % on the challenging AlpacaEval2 dataset. Additional details regarding our experimental setup are available in Appendix H.1.

## 6. Analyses

This section provides empirical analyses of the three factors identified in Section 3.

### 6.1. Retriever optimization objective

**Experimental setting.** Iterative preference optimization is performed on LLMs using the different learning objectives outlined in Section 3.1. Alignment experiments are conducted using the Gemma2-2b-it (Team et al., 2024b) and Mistral-7b-it (Jiang et al., 2023a) models, trained on the Ultrafeedback dataset (Cui et al., 2024). Following the methodology of (Dong et al., 2024), we conduct three iterations of training and report the performance of the final checkpoint in Table 3. Model evaluations are performed on AlpacaEval2 (Dubois et al., 2024) and MixEval (Ni et al., 2024). Detailed settings can be found in Appendix H.2.

*Table 3.* Preference optimization objective study on AlpacaEval2 and MixEval. SFT corresponds to the initial chat model.

| | Method | AlpacaEval 2 | | MixEval | MixEval-Hard |
|---|---|---|---|---|---|
| | | LC Winrate | Winrate | Score | Score |
| Gemma2-2b-it | SFT | 36.39 | 38.26 | 0.6545 | 0.2980 |
| | pairwise | 41.39 | 54.60 | 0.6740 | 0.3375 |
| | contrastive | 43.41 | 56.83 | 0.6745 | 0.3315 |
| | ListMLE | **49.77** | **62.05** | 0.6715 | **0.3560** |
| | LambdaRank | 43.76 | 60.56 | **0.6750** | **0.3560** |
| Mistral-7b-it | SFT | 21.14 | 14.22 | 0.7070 | 0.3610 |
| | pairwise | 36.43 | 41.86 | 0.7175 | 0.4105 |
| | contrastive | 38.44 | 42.61 | 0.7260 | 0.4340 |
| | ListMLE | 38.02 | 43.03 | 0.7360 | 0.4200 |
| | LambdaRank | **40.29** | **46.21** | **0.7370** | **0.4400** |

**Observation.** Table 3 presents the results, from which we make the following observations: (1) Contrastive optimization generally outperforms pairwise optimization (*e.g.*, DPO), likely due to its ability to incorporate more negative examples during each learning step. (2) Listwise optimization methods, including ListMLE and LambdaRank, generally demonstrate superior performance compared to both pairwise and contrastive approaches. This is attributed to their utilization of a more comprehensive set of preference information within the candidate list.

### 6.2. Hard negatives

**Experimental setting.** The Mathstral-7b-it model is trained on the GSM8k training set and evaluated its performance on the GSM8k test set. Iterative DPO is employed as the RLHF method, with the gold or correct response

designated as the positive example. The impact of different hard negative variants is investigated, as described in Section 3.2, with the results presented in Figure 4(a). Additionally, the influence of temperature on negative hardness with Lambdarank objective are examined using experiments on the AlpacaEval 2 dataset, with results shown in Figure 4(b). Detailed settings are in Appendix H.5 and H.6.

**Observation.** Figure 4(a) illustrates that the effectiveness of the final LLM is directly correlated with the hardness of the negatives used during training. Harder negatives consistently lead to a more performant LLM. Figure 4(b) further demonstrates that, within a specific range, lower temperatures generate harder negatives, resulting in a more effective final trained LLM. However, much lower temperature could lead to less diverse responses and finally lead to LLM alignment performance drop.

### 6.3. Candidate List

**Experimental setting.** To investigate the impact of inclusiveness and memorization on LLM alignment, experiments are conducted using Gemma2-2b-it, employing the same training settings as in our objective study. For the inclusiveness study, the performance of the trained LLM is evaluated using varying numbers of candidates in the list. For the memorization study, three approaches are compared: (i) using only the current iteration's responses, (ii) using responses from the current and previous iteration, and (iii) using responses from the current and all previous iterations. Detailed settings can be found in Appendix H.7 and H.3.

*Table 4.* Candidate list study with $\mathcal{L}_{\text{pair}}$ on Gemma2-2b-it. Previous iteration responses enhance performance.

| | Alpaca Eval 2 | |
|---|---|---|
| Method | LC Winrate | Winrate |
| SFT | 47.03 | 48.38 |
| Alignment (w. current) | 55.06 | 66.56 |
| Alignment (w. current + prev) | 55.62 | 70.92 |
| Alignment (w. current + all prev) | 56.02 | 72.50 |

**Observation.** Figure 4(c) illustrates the significant impact of candidate list size on LLM alignment performance. As the candidate list size increases, performance improves, albeit with a diminishing rate of return. This is intuitive, given that a bigger candidate list size can contribute to more hard negatives and potentially benefit the model learning (Qu et al., 2020). Table 4 demonstrates that incorporating responses from previous iterations can enhance performance. This is potentially because introducing previous responses can make the candidate list more comprehensive and lead to better preference signal capturing. More explanations are in Appendix H.3.

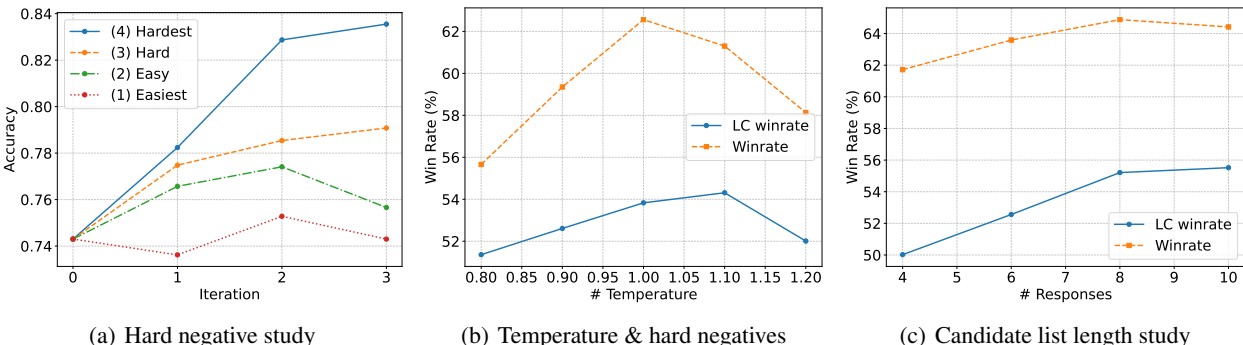

(a) Hard negative study      (b) Temperature & hard negatives      (c) Candidate list length study

*Figure 4.* Hard negative and candidate list study. (a) Hard negative study with $\mathcal{L}_{\text{pair}}$ on GSM8K with Mathstral-7b-it model. We explore four negative settings: (1) a random response not related to the given prompt; (2) a response to a related prompt; (3) an incorrect response to the given prompt with high temperature; (4) an incorrect response to the given prompt with suitable temperature. Hardness: (4)>(3)>(2)>(1). The harder the negatives are, the stronger the trained LLM is. (b) Training temperature study with $\mathcal{L}_{\text{pair}}$ on Mistral-7b-it and Alpaca Eval 2. Within a specific range ($> 1$), lower temperature leads to harder negative and benefit the trained LLM. However, much lower temperature could lead to less diverse responses and finally lead to LLM alignment performance drop. (c) Candidate list size study with $\mathcal{L}_{\text{con}}$ on Mistral-7b-it. As the candidate list size increases, alignment performance improves.

## 7. Related works

**LLM alignment.** Pretrained LLMs demonstrate remarkable capabilities across a broad spectrum of tasks (Brown et al., 2020). Their performance at downstream tasks, such as conversational modeling, is significantly enhanced through alignment with human preferences (Ouyang et al., 2022; Bai et al., 2022). RLHF (Christiano et al., 2017) has emerged as a foundational framework for this alignment, typically involving learning a reward function via a preference model, often using the Bradley-Terry model (Bradley & Terry, 1952), and tuning the LLM using reinforcement learning (RL) to optimize this reward. Despite its success, RLHF's practical implementation is notoriously complex, requiring multiple LLMs, careful hyperparameter tuning, and navigating challenging optimization landscapes.

Recent research has focused on simplifying this process. A line of works studies the direct alignment algorithms (Zhao et al., 2023b; Rafailov et al., 2024; Azar et al., 2024), which directly optimize the LLM in a supervised manner without first constructing a separate reward model. In particular, the representative DPO (Rafailov et al., 2024) attracts significant attention in both academia and industry. After these, SimPO (Meng et al., 2024b) simplifies DPO by using length regularization in place of a reference model.

Although LLMs are adopted for IR (Tay et al., 2022), there is a lack of study to improve direct LLM alignment with IR principles. This paper fills this gap by establishing a systematic link between LLM alignment and IR methodologies, and introducing a novel iterative LLM alignment approach that leverages insights from retriever optimization to advance the state of the art. The most related work is LiPO (Liu et al., 2024), which applies learning-to-rank objectives. However, LiPO relies on off-the-shelf listwise preference

data, which is hard to satisfy in practice.

**Language models for information retrieval.** Language models (LMs) have become integral to modern IR systems (Zhu et al., 2023), particularly after the advent of pretrained models like BERT (Devlin, 2019). A typical IR pipeline employs retrievers and rerankers, often based on dual-encoder and cross-encoder architectures, respectively (Humeau, 2019). Dense Passage Retrieval (DPR) (Karpukhin et al., 2020) pioneered the concept of dense retrieval, laying the groundwork for subsequent research. Building on DPR, studies have emphasized the importance of hard negatives in training (Zhan et al., 2021; Qu et al., 2020) and the benefits of online retriever optimization (Xiong et al., 2020).

In the realm of reranking, Nogueira & Cho (2019) were among the first to leverage pretrained language models for improved passage ranking. This was followed by MonoT5 (Nogueira et al., 2020), which scaled rerankers using large encoder-decoder transformer architectures, and RankT5 (Zhuang et al., 2023), which introduced pairwise and listwise ranking objectives. Recent work has also highlighted the importance of candidate list preprocessing before reranking (Meng et al., 2024a).

Despite the pervasive use of LMs in IR, the interplay between LLM alignment and IR paradigms remains largely unexplored. This work aims to bridge this gap, establishing a strong connection between LLM alignment and IR, and leveraging insights from both fields to advance our understanding of LLM alignment from an IR perspective.

## 8. Conclusions

This paper has forged a novel link between LLM alignment and IR, offering a systematic framework to enhance the LLM alignment performance. Expanding upon this basis, we introduced LᴀʀPO, a new direct preference optimization method that integrates the IR principles to significantly enhance alignment quality. The effectiveness of LᴀʀPO is strongly supported by our comprehensive experiments across widely-used benchmarks, demonstrating its potential as a significant advancement in LLM alignment. Furthermore, our IR-focused analysis highlights the crucial role of retriever optimization objectives, hard negatives, and candidate list construction in achieving effective alignment.

## Acknowledgements

This research was supported in part by Apple PhD Fellowship, in part by US DARPA INCAS Program No. HR0011-21-C0165 and BRIES Program No. HR0011-24-3-0325, in part by the Office of Naval Research contract number N000142412612, in part by NSF grant numbers IIS-19-56151 and 2402873, in part by the Molecule Maker Lab Institute: An AI Research Institutes program supported by NSF under Award No. 2019897 and the Institute for Geospatial Understanding through an Integrative Discovery Environment (I-GUIDE) by NSF under Award No. 2118329, in part by Cisco, and in part by the Center for Intelligent Information Retrieval. Any opinions, findings, and conclusions or recommendations expressed herein are those of the authors and do not necessarily represent the views, either expressed or implied, of the sponsors or the U.S. Government.

## Impact Statement

This paper contributes to the advancement of machine learning. While our work may have broader societal implications, we do not believe any specific impacts warrant explicit discussion in this context.

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

## A. LLM inference strategy and IR pipelines

*Table 5.* Correspondence between LLM inference and IR pipelines.

| Method | Retriever | Reranker | Pipeline |
|---|---|---|---|
| Greedy decoding | LLM | $\varnothing$ | Retriever-only |
| Best-of-N (Stiennon et al., 2020) | LLM | Reward model | Retriever-reranker |
| Majority voting (Wang et al., 2022) | LLM | Majority | Retriever-reranker |
| Iterative refinement (Madaan et al., 2024) | LLM | $\varnothing$ | Iterative retrieval w. query rewriting |

## B. How can SFT and preference optimization help the LLM from an IR perspective?

We assess how well LLMs perform at two tasks: fine-grained reranking (using greedy decoding accuracy) and coarse-grained retrieval (using Recall@$N$). We focus on how SFT and DPO, affect these abilities. Using the Mistral-7b model, we evaluate on the GSM8k and MATH datasets with two approaches: SFT-only, and SFT followed by DPO (SFT $\rightarrow$ DPO).

In the SFT phase, the model is trained directly on correct answers. For DPO, we generate 20 responses per prompt and created preference pairs by randomly selecting one correct and one incorrect response. We use hyperparameter tuning and early stopping to find the best model checkpoints (see Appendix H.4 for details).

*Table 6.* Retrieval (Recall@N) and reranking (greedy accuracy) metrics across dataset and training strategies, with Mistral-7b as the LLM. 0.7 is used as the temperature. Recall@N can also be denoted as pass@N.

| | Metric | init model | SFT | SFT $\rightarrow$ DPO |
|---|---|---|---|---|
| GSM8K | Greedy Acc | 0.4663 | 0.7680 | 0.7991 |
| | Recall@20 | 0.8347 | 0.9462 | 0.9545 |
| | Recall@50 | 0.9090 | 0.9629 | 0.9727 |
| | Recall@100 | 0.9477 | 0.9735 | 0.9826 |
| Math | Greedy Acc | 0.1004 | 0.2334 | 0.2502 |
| | Recall@20 | 0.2600 | 0.5340 | 0.5416 |
| | Recall@50 | 0.3354 | 0.6190 | 0.6258 |
| | Recall@100 | 0.4036 | 0.6780 | 0.6846 |

The results are shown in Table 6. We observe that both SFT and DPO improve both retrieval and reranking, with SFT being more effective. Adding DPO after SFT further improves performance on both tasks. This is consistent with information retrieval principles that both direct retriever optimization and reranker-retrieval distillation can enhance the retriever performance, while the latter on top of the former can further improve the performance. Further discussions can be found in Appendices C and D.

## C. Discussion on the connection and difference between SFT and direct retriever optimization

As discussed in Section 2.3, the direct retriever optimization goal with InfoNCE is shown as:

$$\max \log P(d_{\text{gold}}|q) = \max \log \frac{\text{Enc}_d(d_{\text{gold}}) \cdot \text{Enc}_q(q)}{\sum_{j=1}^{|C|} \text{Enc}_d(d_j) \cdot \text{Enc}_q(q)},$$

while the SFT optimization goal is shown as:

$$\max \log P(y_{\text{gold}}|x) = \max \log \prod_i^{|y_{\text{gold}}|} P(y_{\text{gold}}(i)|z_i) = \max \sum_i^{|y_{\text{gold}}|} \log \frac{\text{Emb}(y_{\text{gold}}(i)) \cdot \text{LLM}(z_i)}{\sum_{j=1}^{|V|} \text{Emb}(v_j) \cdot \text{LLM}(z_i)}. \quad (8)$$

As a result, the SFT objective can be seen as a summation of multiple retrieval optimization objectives, where LLM$(\cdot)$ and word embedding Emb$(\cdot)$ are query encoder and passage encoder respectively.

However, for direct retriever optimization with InfoNCE, $\text{Enc}_d(\cdot)$ is usually a large-scale pretrained language model which is computationally expensive on both time and memory. In this case, it is unrealistic to calculate the $\text{Enc}_d(d_j)$ for all $d_j \in C$, when $C$ is large, because of the time constrain and GPU memory constrain. As a result, a widely-adopted technique is to adopt "in-batch negatives" with "hard negatives" to estimate the $\log P(d_{\text{gold}}|q)$ function:

$$\max \log P(d_{\text{gold}}|q) = \max \log \frac{\text{Enc}_d(d_{\text{gold}}) \cdot \text{Enc}_q(q)}{\sum_{j=1}^{|C|} \text{Enc}_d(d_j) \cdot \text{Enc}_q(q)} \sim \max \log \frac{\text{Enc}_d(d_{\text{gold}}) \cdot \text{Enc}_q(q)}{\sum_{i=1}^{|B|} \text{Enc}_d(d_i) \cdot \text{Enc}_q(q) + \sum_{j=1}^{|H|} \text{Enc}_d(d_j) \cdot \text{Enc}_q(q)},$$

where $B$ is the in-batch negative set and $H$ is the hard negative set. Note that $B \bigcup H \subset C$. This objective is more efficient to optimize but is not the original optimization goal. As a result, the learned model after direct retriever optimization is not optimal. It is also found that the hard negatives $H$ is the key to estimate the original optimization goal (Zhan et al., 2021). Thus, reranker-retriever distillation can further improve the retriever by introducing more hard negatives.

On the other hand, LLM optimization, as shown in Eq. (8), can be seen as a summation of multiple retrieval optimization function. In each retrieval step, the passage can be seen as a token and the corpus is the vocabulary space $V$. Given that the passage encoder $\text{Emb}(\cdot)$ (word embedding) here is cheap to compute and the vocabulary space $V$ ($<100$k) is usually not as large as $C$ ($>$1M) in IR, the objective in Eq. (8) can be directly optimized without any estimation. In this case, the LLM as a retriever is more sufficiently trained compared with the retriever training in IR.

## D. Discussion on the connection and difference between preference optimization and reranker-retriever distillation

As discussed in Section 2.3, preference optimization with an online reward model $f_{\text{reward-model}}(\cdot) \xrightarrow{r} \text{data} \xrightarrow{g(\cdot)} f_{\text{LLM}}(\cdot)$ can be seen as a reranker to retriever distillation process $f_{\text{rerank}}(\cdot) \xrightarrow{r} \text{data} \xrightarrow{g(\cdot)} f_{\text{retrieval}}(\cdot)$, where the reward model is the reranker (*i.e.*, cross-encoder) and the LLM is the retriever (*i.e.*, bi-encoder).

However, there are two slight differences here:

- The LLM after SFT is more sufficiently trained compared to a retriever after direct optimization. As discussed in Appendix C, the SFT optimization function is not an estimated retriever optimization goal compared with the direct retrieval optimization. As a result, the LLM after SFT is suffienctly trained. In this case, if the reward model (reranker) cannot provide information other than that already in the SFT set (*e.g.*, using the SFT prompts), this step may not contribute to significant LLM capability improvement.

- The reward model may introduce auxiliary information than the reranker in IR. For a reranker in IR, it captures a same semantic with the retriever: semantic similarity between the query and the passage. However, in LLM post-training, the goal and data in SFT and preference optimization can be different. For example, the SFT phase could have query/response pairs which enable basic chat-based retrieval capability for the LLM. While the reward model may contain some style preference information or safety information which do not exist in SFT data. In this case, the preference optimization which is the reranker to retriever distillation step could also contribution to performance improvement.

## E. Evaluate LLMs as retrievers

In addition to Mathstral-7b-it on GSM8K in Figure 2, we conduct extensive experiments to both Mistral-7b-it and Mathstral-7b-it on GSM8K and MATH. The results are shown in Figure 5. We have similar findings as in Figure 2 that: (1) As $N$ increases, Recall@$N$ improves significantly, indicating that retrieving a larger number of documents increases the likelihood of including a correct one within the set. (2) For smaller values of $N$ (e.g., $N = 1$), lower temperatures yield higher Recall@$N$. This is because lower temperatures reduce response randomness, favoring the selection of the most relevant result. (3) Conversely, for larger $N$ (e.g., $N > 10$), higher temperatures enhance Recall@$N$. Increased temperature promotes greater response diversity, which, when combined with a larger retrieval set, improves the chances of capturing the correct answer within the results.

## F. LARPO retriever optimization objective

We provide the proof for different variants of LARPO's objective functions.

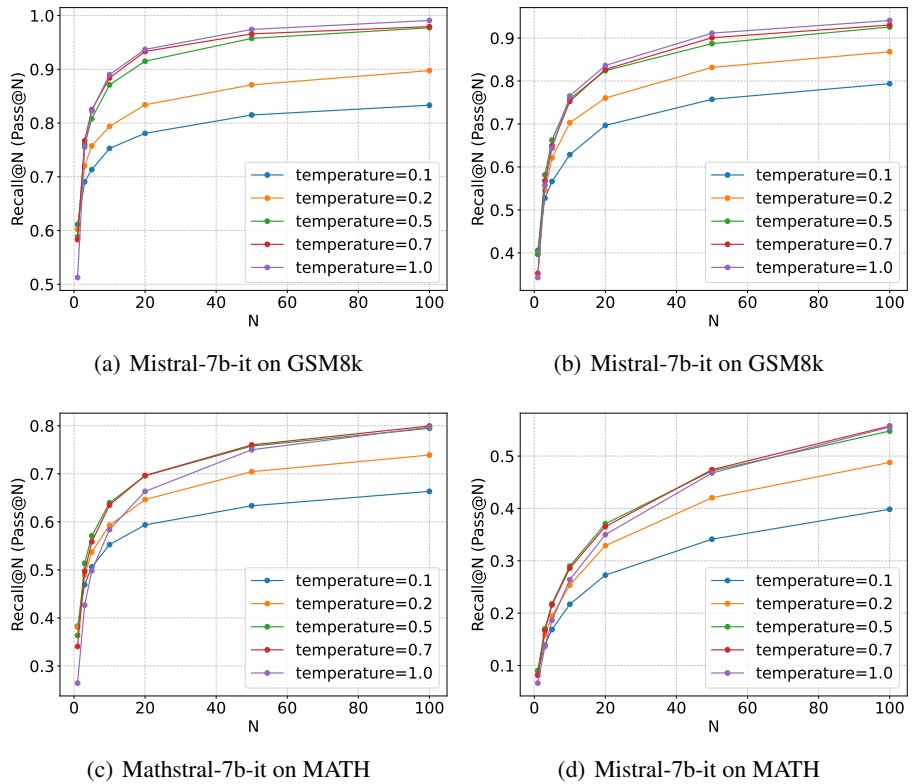

*Figure 5.* Evaluate the LLM as a retriever with Recall@N (Pass@N). As the number (N) of retrieved responses increases, the retrieval recall increases. The higher the temperature is, the broader spectrum the retrieved responses are, and thus the higher the recall is.

### F.1. Contrastive ranking

**Theorem F.1.** *Let $x$ be a prompt and $(y_w, y_l^{(1)}, ..., y_l^{(m)})$ be the responses for $x$ under the contrastive assumption (Eq.(5)). Then the objective function to learn the LLM $\pi_\theta$:*

$$\mathcal{L}_{\text{con}} = -\mathbb{E}\left[\log \frac{\exp\big(\gamma(y_w \mid x)\big)}{\exp\big(\gamma(y_w \mid x)\big) + \sum_{i=1}^{m} \exp\big(\gamma(y_l^{(i)} \mid x)\big)}\right],$$
$$\text{where} \quad \gamma(y \mid x) = \beta \log \frac{\pi_\theta(y \mid x)}{\pi_{\text{ref}}(y \mid x)}. \tag{9}$$

*Proof.* From (Rafailov et al., 2024), we know that

$$r(x, y) = \beta \log \frac{\pi_{\text{llm}}(y|x)}{\pi_{\text{ref}}(y|x)} + \beta \log Z, \tag{10}$$

where $Z = \sum_{y'} \pi_{\text{ref}}(y'|x)\exp(\frac{1}{\beta}r(x, y'))$.

Then,

$$
\begin{aligned}
\Pr(y_w \geq y_l^{(1)}, ..., y_w \geq y_l^{(m)}) &= \mathrm{softmax}(r(x, y_w)) \\
&= \frac{\exp(r(x, y_w))}{\exp(r(x, y_w)) + \sum_{i=1}^{m} \exp(r(x, y_l^{(i)}))} \\
&= \frac{1}{1 + \sum_{i=1}^{m} \exp(r(x, y_l^{(i)}) - r(x, y_w))} \\
&= \frac{1}{1 + \sum_{i=1}^{m} \exp(\gamma(y_l^{(i)} \mid x) + \beta \log Z - \gamma(y_w \mid x) - \beta \log Z)} \\
&= \frac{1}{1 + \sum_{i=1}^{m} \exp(\gamma(y_l^{(i)} \mid x) - \gamma(y_w \mid x))} \\
&= \frac{\exp\big(\gamma(y_w \mid x)\big)}{\exp\big(\gamma(y_w \mid x)\big) + \sum_{i=1}^{m} \exp\big(\gamma(y_l^{(i)} \mid x)\big)}
\end{aligned}
\tag{11}
$$

We can learn $\pi_\theta$ by maximizing the logarithm-likelihood:

$$
\max \log \Pr(y_w \geq y_l^{(1)}, \ldots, y_w \geq y_l^{(m)}) \Leftrightarrow \min -\log \Pr(y_w \geq y_l^{(1)}, \ldots, y_w \geq y_l^{(m)}) = \mathcal{L},
\tag{12}
$$

$$
\therefore \mathcal{L}_{\mathrm{con}} = -\mathbb{E}\left[ \log \frac{\exp\big(\gamma(y_w \mid x)\big)}{\exp\big(\gamma(y_w \mid x)\big) + \sum_{i=1}^{m} \exp\big(\gamma(y_l^{(i)} \mid x)\big)} \right],
\tag{13}
$$

$$
\text{where} \quad \gamma(y \mid x) = \beta \log \frac{\pi_\theta(y \mid x)}{\pi_{\mathrm{ref}}(y \mid x)}.
\tag{14}
$$

### F.2. LambdaRank ranking

**Theorem F.2.** *Let $x$ be a prompt and $(y_1, ..., y_m)$ be the responses for $x$ under the LambdaRank assumption (Eq.(6)). Then the objective function to learn the LLM $\pi_\theta$:*

$$
\mathcal{L}_{\mathrm{lamb}} = -\mathbb{E}\left[ \sum_{1 < i < j < m} \log \sigma\Big( \gamma(y_i \mid x) - \gamma(y_j \mid x) \Big) \right].
\tag{15}
$$

*Proof.*

$$
\begin{aligned}
\Pr(y_1 \geq ... \geq y_m) &= \prod_{1 < i < j < m} \sigma(r(x, y_i) - r(x, y_j)) \\
&= \prod_{1 < i < j < m} \sigma(\gamma(x, y_i) + \beta \log Z - \gamma(x, y_j) - \beta \log Z) \\
&= \prod_{1 < i < j < m} \sigma(\gamma(y_i \mid x) - \gamma(y_j \mid x)).
\end{aligned}
\tag{16}
$$

We can learn $\pi_\theta$ by maximizing the logarithm-likelihood:

$$
\max \log \Pr(y_w \geq y_l^{(1)}, \ldots, y_w \geq y_l^{(m)}) \Leftrightarrow \min -\log \Pr(y_w \geq y_l^{(1)}, \ldots, y_w \geq y_l^{(m)}) = \mathcal{L},
\tag{17}
$$

$$
\therefore \mathcal{L}_{\mathrm{lamb}} = -\mathbb{E}\left[ \sum_{1 < i < j < m} \log \sigma\Big( \gamma(y_i \mid x) - \gamma(y_j \mid x) \Big) \right],
\tag{18}
$$

$$
\text{where} \quad \gamma(y \mid x) = \beta \log \frac{\pi_\theta(y \mid x)}{\pi_{\mathrm{ref}}(y \mid x)}.
\tag{19}
$$

### F.3. ListMLE ranking

**Theorem F.3.** *Let $x$ be a prompt and $(y_1, ..., y_m)$ be the responses for $x$ under the ListMLE assumption (Eq.(7)). Then the objective function to learn the LLM $\pi_\theta$:*

$$\mathcal{L}_{\text{lmle}} = -\mathbb{E}\left[\sum_{i=1}^{m} \log \frac{\exp\big(\gamma(y_i \mid x)\big)}{\exp\big(\gamma(y_i \mid x)\big) + \sum_{j=i}^{m} \exp\big(\gamma(y_j \mid x)\big)}\right]. \tag{20}$$

*Proof.* From Eq.(11),

$$\begin{aligned}
\Pr(y_1 \succeq ... \succeq y_m) &= \prod_{i=1}^{m} \Pr(y_i \succeq y_{i+1}, ..., y_i \succeq y_m) \\
&= \prod_{i=1}^{m} \frac{\exp(\gamma(y_i \mid x))}{\exp(\gamma(y_i \mid x)) + \sum_{j=i+1}^{m} \exp(\gamma(y_j \mid x))}
\end{aligned}. \tag{21}$$

We can learn $\pi_\theta$ by maximizing the logarithm-likelihood:

$$\max \log \Pr(y_w \succeq y_l^{(1)}, \ldots, y_w \succeq y_l^{(m)}) \Leftrightarrow \min -\log \Pr(y_w \succeq y_l^{(1)}, \ldots, y_w \succeq y_l^{(m)}) = \mathcal{L}, \tag{22}$$

$$\therefore \mathcal{L}_{\text{lmle}} = -\mathbb{E}\left[\sum_{i=1}^{m} \log \frac{\exp\big(\gamma(y_i \mid x)\big)}{\exp\big(\gamma(y_i \mid x)\big) + \sum_{j=i+1}^{m} \exp\big(\gamma(y_j \mid x)\big)}\right], \tag{23}$$

$$\text{where} \quad \gamma(y \mid x) = \beta \log \frac{\pi_\theta(y \mid x)}{\pi_{\text{ref}}(y \mid x)}. \tag{24}$$

## G. Baselines

We conduct detailed illustrations on the baselines compared with LARPO in Section 5 below.

- RRHF (Yuan et al., 2023) scores responses via a logarithm of conditional probabilities and learns to align these probabilities with human preferences through ranking loss.

- SLiC-HF (Zhao et al., 2023b) proposes a sequence likelihood calibration method which can learn from human preference data.

- DPO (Guo et al., 2024) simplifies the PPO (Ouyang et al., 2022) algorithms into an offline direct optimization objective with the pairwise Bradley-Terry assumption.

- IPO (Azar et al., 2024) theoretically grounds pairwise assumption in DPO into a pointwise reward.

- CPO (Xu et al., 2024a) adds a reward objective with sequence likelihood along with the SFT objective.

- KTO (Ethayarajh et al., 2024) adopts the Kahneman-Tversky model and proposes a method which directly maximizes the utility of generation instead of the likelihood of the preferences.

- RDPO (Park et al., 2024) modifies DPO by including an additional regularization term to disentangle the influence of length.

- SimPO (Meng et al., 2024b) further simplifies the DPO objective by using the average log probability of a sequence as the implicit reward and adding a target reward margin to the Bradley-Terry objective.

- Iterative DPO (Xiong et al., 2024) identifies the challenge of offline preference optimization and proposes an iterative learning framework.

## H. Experiment settings

### H.1. Table 2

We conduct evaluation on two widely used benchmark: AlpacaEval2 (Dubois et al., 2024) and MixEval (Ni et al., 2024). We consider two base models: Mistral-7b-base and Mistral-7b-it. For Mistral-7b-base, we first conduct supervised finetuning following Meng et al. (2024b) before the preference optimization.

*Table 7.* Preference optimization objective study on AlpacaEval2 and MixEval. For AlpacaEval2, we report the result with both opensource LLM evaluator `alpaca_eval_llama3_70b_fn` and GPT4 evaluator `alpaca_eval_gpt4_turbo_fn`. SFT corresponds to the initial chat model.

| | Method | AlpacaEval 2 (opensource LLM) | | AlpacaEval 2 (GPT-4) | | MixEval | MixEval-Hard |
|---|---|---|---|---|---|---|---|
| | | LC Winrate | Winrate | LC Winrate | Winrate | Score | Score |
| Gemma2-2b-it | SFT | 47.03 | 48.38 | 36.39 | 38.26 | 0.6545 | 0.2980 |
| | pairwise | 55.06 | 66.56 | 41.39 | 54.60 | 0.6740 | 0.3375 |
| | contrastive | 60.44 | 72.35 | 43.41 | 56.83 | 0.6745 | 0.3315 |
| | ListMLE | 63.05 | 76.09 | 49.77 | 62.05 | 0.6715 | 0.3560 |
| | LambdaRank | 58.73 | 74.09 | 43.76 | 60.56 | 0.6750 | 0.3560 |
| Mistral-7b-it | SFT | 27.04 | 17.41 | 21.14 | 14.22 | 0.7070 | 0.3610 |
| | pairwise | 49.75 | 55.07 | 36.43 | 41.86 | 0.7175 | 0.4105 |
| | contrastive | 52.03 | 60.15 | 38.44 | 42.61 | 0.7260 | 0.4340 |
| | ListMLE | 48.84 | 56.73 | 38.02 | 43.03 | 0.7360 | 0.4200 |
| | LambdaRank | 51.98 | 59.73 | 40.29 | 46.21 | 0.7370 | 0.4400 |

The performance scores for offline preference optimization baselines are from SimPO (Meng et al., 2024b). To have a fair comparison with these baselines, we adopt the same off-the-shelf reward model (Jiang et al., 2023b) as in SimPO for the iterative DPO baseline and LARPO.

For the iterative DPO baseline, we generate 2 responses for each prompt, score them with the off-the-shelf reward model and construct the preference pair data to tune the model.

For LARPO (contrastive $\mathcal{L}_{con}$), we generate 10 responses each iteration and score them with the reward model. The top-1 ranked response and the bottom-3 ranked responses are adopted as the chose response and rejected responses respectively. Generation temperature is selected as 1 and 0.8 for Mistral-7b-base and Mistral-7b-it respectively (we search it among 0.8, 0.9, 1.0, 1.1, 1.2).

For LARPO (LambdaRank $\mathcal{L}_{lamb}$), we generate 10 responses each iteration and score them with the reward model. The top-2 ranked response and the bottom-2 ranked responses are adopted as the chose response and rejected responses respectively. Generation temperature is selected as 1 and 0.8 for Mistral-7b-base and Mistral-7b-it respectively (we search it among 0.8, 0.9, 1.0, 1.1, 1.2).

For LARPO (ListMLE $\mathcal{L}_{lmle}$), we generate 10 responses each iteration and score them with the reward model. The top-2 ranked response and the bottom-2 ranked responses are adopted as the chose response and rejected responses respectively. Generation temperature is selected as 1 and 0.8 for Mistral-7b-base and Mistral-7b-it respectively (we search it among 0.8, 0.9, 1.0, 1.1, 1.2).

LARPO can achieve even stronger performance with stronger off-the-shelf reward model (Dong et al., 2024).

## H.2. Table 3

We conduct experiments on both Gemma2-2b-it (Team et al., 2024b) and Mistral-7b-it (Jiang et al., 2023a). Following Tunstall et al. and Dong et al. (2024), we perform training on UltraFeedback dataset for 3 iterations and show the performance of the final model checkpoint. We use the pretrained reward model from Dong et al. (2024). The learning rate is set as 5e-7 and we train the LLM for 2 epochs per iteration.

For the pairwise objective, we generate 2 responses for each prompt and construct the preference pair data with the reward model. For the others, we generate 4 responses per prompt and rank them with the reward model. For the contrastive objective, we construct the 1-vs-N data with the top-1 ranked response and the other responses. For the listMLE and lambdarank objective, we take the top-2 as positives and the last-2 as the negatives. Experiments with opensource LLM as the evaluator (`alpaca_eval_llama3_70b_fn`) can be found in Table 7.

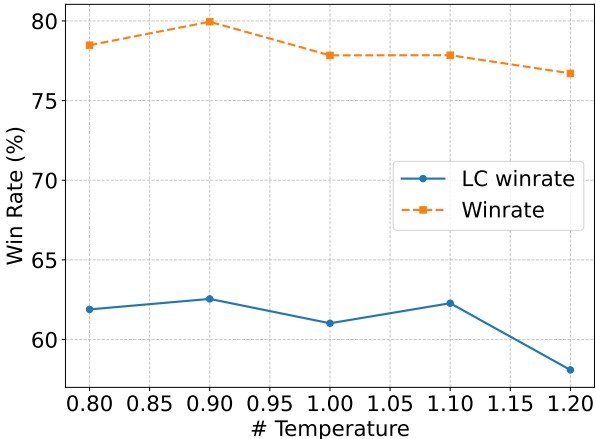

*Figure 6.* Training temperature study with $\mathcal{L}_{\text{pair}}$ on Gemma2-2b-it and Alpaca Eval 2. Within a specific range ($> 0.9$), lower temperature leads to harder negative and benefit the trained LLM. However, temperature lower than this range can cause preferred and rejected responses non-distinguishable and lead to degrade training.

### H.3. Table 4

We adopt Gemma2-2b-it as the initial model. All the models are trained with iterative DPO for 3 iterations. We use the off-the-shelf reward model (Dong et al., 2024). We generate 2 responses for each prompt in each iteration. For "w. current", we only use the scored responses in the current iteration for preference optimization data construction. For "w. current + prev", we rank the responses in the current iteration and the previous one iteration, and construct the preference pair data with the top-1 and bottom-1 ranked responses. For "w. current + all prev", we rank all the responses for the prompt in the current and previous iterations and construct the preference pair data. For "single temperature", we only adopt temperature 1 and generate 2 responses for reward model scoring. For "diverse temperature", we generate 2 responses with temperature 1 and 0.5 respective and rank the 4 responses to construct the preference data with the reward model.

### H.4. Table 6

We use mistral-7b-it (Jiang et al., 2023a) as the initial model to alleviate the influence of the math related post-training data of the original model. For SFT, we conduct training on the meta-math dataset (Yu et al., 2023). For DPO, we use the prompts in the training set of the two dataset and conduct online iterative preference optimization with the binary rule-based reward (measure if the final answer is correct or not with string match). The evaluation is performed on the test set of MATH and GSM8K respectively. For SFT, we follow the same training setting with Yu et al. (2023). For DPO, we search the learning rate in 1e-7, 2e-7, 5e-7, 2e-8, 5e-8 and train the LLM for 5 iterations with early stop (1 epoch per iteration for MATH and 2 epoch per iteration for GSM8K). The learning rate is set as 1e-7 and we select the checkpoint after the first and fourth iteration for GSM8K and MATH respectively.

### H.5. Figure 4(a)

We conduct training with the prompts in the training set of GSM8K and perform evaluation on GSM8K testing set. We conduct learning rate search and finalize it to be 2e-7. The learning is performed for 3 iterations.

We make explanations of how we construct the four types of negative settings: For (1) a random response not related to the given prompt, we select a response for a random prompt in Ultrafeedback. For (2) a response to a related prompt, we pick up a response for a different prompt in the GSM8K training set. For (3) an incorrect response to the given prompt with high temperature, we select the temperature to be 1. For (4) an incorrect response to the given prompt with low temperature, we select the temperature to be 0.7.

### H.6. Figure 4(b)

We conduct experiments on both Gemma2-2b-it and Mistral-7B-it models. For both LLMs, we conduct iterative DPO for 3 iterations and report the performance of the final model. We perform evaluation on Alpaca Eval2 with

`alpaca_eval_llama3_70b_fn` as the evaluator.

For temperature study, we find that under a specific temperature threshold, repeatedly generated responses will be large identical for all LLMs and cannot be used to construct preference data, while the threshold varies for different LLMs. The "low" and "high" refer to the value of those selected temperatures. We also conduct experiments on Gemma2-2b-it model and show the results in Figure 6.

### H.7. Figure 4(c)

We adopt Mistral-7b-it as the initial LLM and the contrastive objective (Eq. 9) in iterative preference optimization. We generate 4/6/8/10 responses with the LLM and score the responses with the off-the-shelf reward model (Dong et al., 2024). The top-1 scored response is adopted as the positive response and the other responses are treated as the negative responses to construct the 1-vs-N training data. The temperature is set as 1 to generate the responses.

