# OpenReview forum: "LLM Alignment as Retriever Optimization: An Information Retrieval Perspective"
_ICML.cc/2025/Conference — ICML 2025 poster_

### Official Review · Reviewer_ToBQ · 2025-03-09

**Overall Recommendation:** 4

**Summary:**

This paper establishes the connections between the formulations of LLM alignment and information retrieval (IR). Inspired by this discovery, it introduces various practices in information retrieval into LLM alignment, including hard negative mining and ranking loss functions. Empirical studies demonstrate the effectiveness of the proposed approach.

## update after rebuttal

The additional empirical results and conceptual explanations by the authors have mostly addressed my earlier concerns.

**Claims And Evidence:**

The claims are mostly supported by clear and convincing evidence.

**Essential References Not Discussed:**

N.A.

**Experimental Designs Or Analyses:**

I checked the experimental designs and analyses.

Possible issues include:

**I1.** The authors only compared the baseline approaches with the weaker reward model, and it is unclear how the baseline methods perform with the stronger reward model.

**I2.** The experiments are restricted to small models with 2B or 7B parameters.

**I3.** SimPO also studies two Llama-3 models in addition.

**Methods And Evaluation Criteria:**

The proposed methods and/or evaluation criteria (e.g., benchmark datasets) make sense for the problem or application at hand.

**Other Comments Or Suggestions:**

N.A.

**Other Strengths And Weaknesses:**

### Other Strengths

**S1.** The paper is well-written and easy to follow overall.

### Other Weaknesses

**W1.** Notations in section 2 are not self-contained.

**Questions For Authors:**

See above.

**Relation To Broader Scientific Literature:**

To the best of my knowledge,

- The IR perspective is novel.
- The paper considers three IR-inspired objectives, including contrastive ranking and two list-wise ranking objectives (LambdaRank and ListMLE). [1] has previously explored using learning-to-rank objectives. The differences between the objectives employed by [1] and this paper are not sufficiently discussed.
- The empirical studies follow the practice (setting, benchmarks, baselines) of existing efforts [2].
- The mining of hard negatives in iterative alignment is novel.

[1] Liu et al. LiPO: Listwise Preference Optimization through Learning-to-Rank.

[2] Meng et al. SimPO: Simple Preference Optimization with a Reference-Free Reward.

**Theoretical Claims:**

I checked the proofs in Appendix E and did not find issues.

---

> ### Author Rebuttal · Authors · 2025-03-31
>
> We thank the reviewer for the helpful feedback and address the points below:
>
> - **Baselines with Strong Reward Models.** We include iterative DPO with a strong reward model for fair comparison:
>
> | Model | ApEval 2 (LC WR) |  ApEval 2 (WR) | MixEval | MixEval-Hard |
> |--|--|--|--|--|
> | **Mistral-Base (7B)** |
> | Iterative DPO | 32.85 | 35.2 | 0.6825 | 0.3835 |
> | LarPO (Contrastive) | 41.5 | 42.9 | 0.718 | 0.417 |
> | LarPO (LambdaRank) | 35.8 | 34.1 | 0.717 | 0.431 |
> | LarPO (ListMLE) | 36.6 | 37.8 | 0.73 | 0.423 |
> | **Mistral-Instruct (7B)** |
> | Iterative DPO | 38.95 | 47.89 | 0.698 | 0.3965 |
> | LarPO (Contrastive) | 43 | 53.8 | 0.718  | 0.425 |
> | LarPO (LambdaRank) | 41.9 | 48.1 | 0.74 | 0.44 |
> | LarPO (ListMLE) | 39.6 | 48.1  | 0.717 | 0.397 |
>
> LarPO consistently outperforms iterative DPO under the strong reward model, demonstrating the strength of our proposed frameworks.
>
> - **Model Size Limitation.** Training larger models is resource-intensive and cannot be easily handled by our current computational devices. Our current results on 2B and 7B models show LarPO’s consistent effectiveness across these model sizes and families. We leave scaling to larger models for future work.
>
> - **LLaMA-3 Results.** We include LLaMA-3-8B model results below. LarPO achieves the best performance across metrics, outperforming all baselines including SimPO:
>
> | Model | ApEval 2 (LC WR)  | ApEval 2 (WR) | Arena-Hard | MTBench | MixEval | MixEval-Hard |
> |--|--|--|--|--|--|--|
> | SFT | 26.0 | 25.3 | 22.3 | 8.1 | 0.742 | 0.4005 |
> | RRHF | 31.3 | 28.4 | 26.5 | 7.9 | 0.743 | 0.4125 |
> | SLiC-HF | 26.9 | 27.5 | 26.2 | 8.1 | 0.752 | 0.4515 |
> | DPO | 40.3 | 37.9 | 32.6 | 8.0 | 0.7715 | 0.4675 |
> | IPO | 35.6 | 35.6 | 30.5 | 8.3 | 0.756 | 0.452 |
> | CPO | 28.9 | 32.2 | 28.8 | 8.0 | 0.7665 | 0.4225 |
> | KTO | 33.1 | 31.8 | 26.4 | 8.2 | 0.7645 | 0.461 |
> | ORPO | 28.5 | 27.4 | 25.8 | 8.0 | 0.7545 | 0.445 |
> | RDPO | 41.1 | 37.8 | 33.1 | 8.0 | 0.779 | 0.4645 |
> | SimPO | 44.7 | 40.5 | 33.8 | 8.0 | 0.732 | 0.4185 |
> | LarPO (Contrastive)| **47.72** | **49.81** | 35.2 | 8.3  | **0.7795** | 0.4555 |
> | LarPO (LambdaRank) | 46.2 | 49.07 | **36.5** | **8.4** | 0.779 | **0.4785** |
> | LarPO (ListMLE) | 44.67 | 47.51 | 35.6 | 8.3 | 0.7705 | 0.456 |
>
>
>
> - **Comparison to LiPO.** LiPO proposes a LambdaRank-based listwise objective, which can be viewed as a special case of LarPO. However, their objective designs are based on learning-to-rank heuristics without a grounded theoretical basis. In contrast, LarPO is supported by theoretical foundations and generalizes to multiple ranking objectives beyond LambdaRank. We will add this distinction in the revised manuscript.
>
> - **W1. Notations in section 2 are not self-contained.** We acknowledge this and will revise to ensure all notations are self-contained and clearly defined.

---

> > ### Comment · Reviewer_ToBQ · 2025-04-01
> >
> > Thank you for the detailed response, which has mostly addressed my concerns. I do not have further questions.

---

> > > ### Author Response · Authors · 2025-04-08
> > >
> > > Dear Reviewer ToBQ,
> > >
> > > Thank you once again for your thoughtful feedback, which has been invaluable in helping us improve our paper. We’re glad to hear that our response has addressed most of your concerns.
> > >
> > > Wishing you all the best.

---

### Official Review · Reviewer_bKqw · 2025-03-10

**Overall Recommendation:** 3

**Summary:**

This paper views LLM alignment as a retriever optimization problem, presenting a systematic framework that connects LLM alignment with information retrieval (IR) methodologies. The paper maps LLM generation and reward models to the retriever-reranker paradigm in IR. Based on three key IR principles—retriever optimization objectives, hard negative mining, and candidate list construction—the authors propose a novel alignment method, LLM Alignment as Retriever Preference Optimization (LARPO). Experimental results on AlpacaEval2 and MixEval-Hard validate the effectiveness of the proposed LARPO approach.

## update after rebuttal

Thanks for the author's rebuttal and additional experiments. I appreciate the authors' conceptualization of LLM tuning as retriever optimization, where SFT is treated as direct retriever optimization and preference optimization is framed as reranker-retriever distillation. While this perspective is intuitive, the paper lacks a clear and convincing explanation of why IR principles are effective for LLM alignment. Unfortunately, this key concern was not addressed in the rebuttal. **I hope the authors can consider discussing this aspect more thoroughly in future revisions. I will raise my score to a 3, but I still retain this concern.**

**Claims And Evidence:**

The authors conceptualize LLM tuning as retriever optimization, where supervised fine-tuning (SFT) is treated as direct retriever optimization and preference optimization is framed as reranker-retriever distillation. While this perspective is intuitive, the paper lacks a clear explanation of why IR principles are effective for LLM alignment.

**Essential References Not Discussed:**

Please see above in Experimental Designs Or Analyses about the following papers:

[1] LiPO: Listwise Preference Optimization through Learning-to-Rank
[2] Optimizing Preference Alignment with Differentiable NDCG Ranking
[3] MPPO: Multi Pair-wise Preference Optimization for LLMs with Arbitrary Negative Samples
[4] Reward difference optimization for sample reweighting in offline rlhf
[5] Not all preference pairs are created equal: A recipe for annotation-efficient iterative preference learning

**Experimental Designs Or Analyses:**

Some experimental comparisons seem unfair. I have the following concerns:

**Lack of Ablation Studies**: While framing LLM alignment as retriever optimization is a valuable perspective, the proposed method combines elements of optimization objectives, hard negative mining, and candidate list construction. Each of these components has been explored in prior work, but the paper does not provide comparisons.

 + For listwise preference optimization, comparisons should be made against existing methods such as LiPO [1], DRPO [2], and MPPO [3].

+ The baseline methods in Table 2 use different training datasets than LARPO. To fairly assess the effectiveness of the proposed preference optimization objective, Table 3 should include comparisons against other DPO-based baselines under the same data conditions. The current results do not convincingly demonstrate that the optimization objective itself is effective, as improvements may stem from external reward model filtering rather than the optimization technique.

+ For hard negatives mining, comparisons should be made with existing preference pair quality assessment techniques, such as explicit reward margin [4] and implicit reward margin [5]. This would provide stronger evidence for the effectiveness of the proposed hard negatives mining approach.

**Temperature Coefficient in Figure 4(b)**: The experiment starts at a temperature of 0.8. It is unclear why lower values (0–0.8) were omitted. A complete trend analysis is necessary to understand the behavior across the full range.

[1] LiPO: Listwise Preference Optimization through Learning-to-Rank
[2] Optimizing Preference Alignment with Differentiable NDCG Ranking
[3] MPPO: Multi Pair-wise Preference Optimization for LLMs with Arbitrary Negative Samples
[4] Reward difference optimization for sample reweighting in offline rlhf
[5] Not all preference pairs are created equal: A recipe for annotation-efficient iterative preference learning

**Methods And Evaluation Criteria:**

1. The authors adopt most of their offline experimental results directly from SimPO but use only the AlpacaEval2 dataset, replacing Arena-Hard and MT-Bench with MixEval. It is unclear why this dataset substitution was made. Given the reliance on SimPO's experimental setup, the authors should clarify why they did not use the same datasets entirely.

2. The paper lacks an analysis of the computational complexity associated with different optimization objectives.

**Other Comments Or Suggestions:**

Please see above.

**Other Strengths And Weaknesses:**

1. The perspective introduced in this paper is valuable and contributes to a conceptual understanding of LLM alignment through the lens of retriever optimization. However, the proposed method primarily integrates existing approaches in optimization objectives, hard negatives mining, and candidate list construction, serving more as an explanatory framework rather than introducing a fundamentally new method. This raises concerns about the validity of the claimed LARPO method, particularly due to the lack of thorough ablation studies to demonstrate its effectiveness. Consequently, while I believe this is a valuable perspective paper, I question whether it is suitable for a main conference paper.

2. The authors claim that the baseline checkpoints are from SimPO. Could you provide the corresponding checkpoint links? Additionally, will the author open-source the code, experimental data, and checkpoints used in this paper, which will enhance the reproducibility?

**Questions For Authors:**

NA

**Relation To Broader Scientific Literature:**

The authors establish a conceptual link between LLM alignment and retriever optimization, leading to the proposal of the LARPO method.

**Theoretical Claims:**

The theoretical section appears to be correct.

---

> ### Author Rebuttal · Authors · 2025-03-31
>
> We appreciate your insightful feedback and believe it has significantly strengthened our manuscript. We have carefully addressed each of your comments as detailed below:
>
> - **Use of SimPO Datasets.** We exclude MTBench and Arena-Hard because (1) SimPO shows minimal differences among methods on MTBench, and (2) Arena-Hard evaluation is costly ($30+ per run). In this rebuttal, we include both below using LLaMA-3 (8B) and Mistral (7B):
>
> Llama-3-8B models
> |Model|Arena-Hard|MTBench|
> |-|-|-|
> |RRHF|26.5|7.9|
> |SLiC-HF|26.2|8.1|
> |DPO|32.6|8.0|
> |IPO|30.5|8.3|
> |CPO|28.8|8.0|
> |KTO|26.4|8.2|
> |RDPO|33.1|8.0|
> |SimPO|33.8|8.0|
> |LarPO (Contrastive)|35.2|8.3|
> |LarPO (LambdaRank)|**36.5**|**8.4**|
> |LarPO (ListMLE)|35.6|8.3|
>
> Mistral-7B models
> |Model|Arena-Hard|MTBench|
> |-|-|-|
> |RRHF|5.8|6.7|
> |SLiC-HF|7.3|**7.4**|
> |DPO|10.4|7.3|
> |IPO|7.5|7.2|
> |CPO|6.9|6.8|
> |KTO|5.6|7.0|
> |RDPO|8.0|**7.4**|
> |SimPO|16.6|7.3|
> |LarPO (Contrastive)|15.4|7.2|
> |LarPO (LambdaRank)|**19.7**|6.9|
> |LarPO (ListMLE)|14.2|7.3|
>
> These results show LarPO performs comparable or better than competitive baselines on these benchmarks. We will include these results in the revised manuscript.
>
> - **Computational Complexity.** (1) Pairwise Ranking: Computes preference between a single pair of responses per prompt, with complexity O(1). (2) Contrastive Ranking: Involves a softmax over a candidate list of size k, with complexity O(k). In practice, good performance is achieved with a small k (e.g., with k = 4, contrastive ranking performs better alpaca_eval2 win rate than pairwise ranking), making it effective. (3) LambdaRank: Based on pairwise comparisons with position-aware weighting. Its worst-case complexity is O(k²) but can be reduced to O(k) via subsampling. (4) ListMLE: Computes the likelihood of a full ranking using sequential softmax, with O(k²) complexity. This can also be lowered to O(k) using subsampling strategies.
>
> - **Comparison to LiPO[1], DRPO[2], MPPO[3], and reward margin[4][5] papers.** We would like to first emphasize that our goal is to connect IR and alignment, not to propose new listwise or hard negative methods. However, we are happy to compare our method to the five mentioned papers and add discussion accordingly in our revision. (1) LiPO [1] proposes LLM alignment as listwise learning to rank and proposes a LambdaRank-based solution. LiPO is a special case of LarPO with LambdaRank. However, we provide extensive theoretical background and demonstrate that many other ranking assumptions can be adopted in LarPO in addition to LambdaRank. (2) We compare LarPO with [2], [4], and [5] in the table below, where LarPO consistently outperforms all methods. Specifically: [2] performs well on the HH dataset (as shown in the original paper) but shows limited effectiveness on Alpaca Eval 2 and MixEval. [4] introduces a reward margin coefficient, which can misguide the LLM if the reward scores lack sufficient granularity. [5] relies on optimizing Eq. 6, but we find its training to be unstable and less robust in our experiments. (3) MPPO [3] is concurrent (posted one month before the ICML deadline), and comparison is left to future work.
>
> |Model|Ap2 (LC WR)|Ap2 (WR)|MixEval|MixEval-Hard|
> |-|-|-|-|-|
> |[2]|9.00|5.11|0.6035|0.2865|
> |[4]|13.48|10.45|0.6785|0.3395|
> |[5]|9.34|6.21|0.6360|0.3285|
> |LarPO (Contrastive)|41.50|42.90|0.7180|0.4170|
> |LarPO (LambdaRank)|35.80|34.10|0.7170|0.4310|
> |LarPO (ListMLE)|36.60|37.80|0.7300|0.4230|
>
> - **Table 2 Dataset Consistency.** All baselines in Table 2 use the same UltraFeedback prompt set and PairRM reward model if needed (as in SimPO); thus, the experiments are in fair condition. To further isolate objective effectiveness, we have added results with advanced reward models (i.e., FsfairX) under the same setup, shown in the response 4 to Reviewer r2Em.
>
> - **Temperature Study (Fig 4b).** We previously started from T=0.8 due to response concerns that lower temperatures might reduce response diversity, potentially leading to lower-quality preference data. In this rebuttal, we add these results, as shown in the table below:
>
> |Temperature|Ap2 (LC Winrate)| Ap2 (Winrate)|
> |-|-|-|
> |0.2|55.47|62.74|
> |0.4|53.71|62.09|
> |0.6|55.45|62.30|
>
> Surprisingly, low temperatures yield strong performance—possibly because the lower the temperature is, the harder the negatives are—suggesting an interesting direction on in-depth temperature analysis for future investigation. We will add this discussion to the revised manuscript.
>
> - **Ablation studies.** Section 6 includes ablations on objectives, hard negatives, and candidate construction. Our work is the first to jointly explore all these components in connecting retrieval and LLM alignment which we would think are enough contributions to the main conference.
>
> - **Reproducibility.** SimPO paper checkpoints: https://huggingface.co/collections/princeton-nlp/simpo-66500741a5a066eb7d445889. We commit to releasing all code, data, and checkpoints upon acceptance.

---

### Official Review · Reviewer_r2Em · 2025-03-12

**Overall Recommendation:** 4

**Summary:**

This paper demonstrates that concepts from information retrieval can be ported over to shed light on numerous aspects of language model alignment tuning, including both RLHF-type functions and the data generation methods that feed those objectives. From a more technical perspective, this paper demonstrates empirically that replacing the Bradley-Terry pairwise preferences model used in the original DPO paper with listwise preference models like LambdaRank can result and improved alignment.

# Update after rebuttal

I appreciate the authors' thorough reply to my questions/comments, and maintain my recommendation that the paper be accepted.

**Claims And Evidence:**

Largely yes. The theoretical discussion (mapping ideas from information retrieval onto LLM alignment, both conceptually and mathematically) is sound, and the experiments are for the most part appropriately designed to measure natural axes of variation motivated by the theoretical discussion. I have the following specific comments/concerns:

Fig 2/Section 2.4: I appreciate the analogy. However:

- I wonder if the experiment could have been done on a single data set (of course 2 different models is unavoidable). E.g. make all answers in the dataset the corpus the retriever is retrieving from.
- Both Recall@N and Pass@N are metrics that obviously monotonically increase with N. So this experiment provides limited support for the analogy between retrieval and LLM inference.
- "Greedy decoding, equivalent to N = 1, is a prevalent LLM inference strategy. However, as shown in Figure 2(b), Pass@1 is often suboptimal". I don't agree with the word choice "suboptimal" since in many real life scenarios you only get to attempt to answer once. And the Pass@N metric just checks whether one of your N attempts was correct, and includes no mechanism for picking which of the N samples to submit as final answer.

**Table 2**: it's a nice sanity check that using a better reward model improves the performance of LARPO, however, the absence of comparison against any baselines (in particular DPO) using the better reward model makes the bottom of this table difficult to interpret.

**Section 6.1**: the observations that list MLE and lambda rank improve over pairwise and contrastive objectives do not seem to transfer to table 2. At least some comment on this would be warranted.

**Figure 4**: could you please comment on why GSM, Mathstral and iterative DPO are used in figure 4a (as they differ from the models/datasets/methods appearing elsewhere in the section)? I do not concur with the analysis of 4b. First, it compares temperature with win rate, not with example hardness. While it is plausible that example hardness is a latent factor and that the impact of temperature and win rate occurs *via* hardness, the experiment and figure simply don't show this. Moreover, when I see this figure I reach the reasonable conclusion that "oh, the simple choice of temperature 1 works best." The analysis "within a specific range, lower
temperatures generate harder negatives" somehow presupposes that the range under consideration is [1,1.2], even though that's not even the entire $x$ axis of the figure.

**Essential References Not Discussed:**

Not that I am aware of.

**Experimental Designs Or Analyses:**

See my comments on claims and evidence.

**Methods And Evaluation Criteria:**

Yes, although as mentioned in the claims and evidence section of this review, I have some questions around consistency in models, datasets and optimization algorithms across the various experiments of the paper.

Note: I'm not asking for more experiments during the rebuttal period. Further clarification in the paper around choice of model and data set and optimization algorithm where they differ from other experiments in the paper would be adequate.

**Other Comments Or Suggestions:**

Equations 1-3: I get what they're trying to say, but it appears odd that the document $d$ does not appear on the right hand side of the set notation. May I suggest either of a) a pseudo-code type `sorted(D, r(q,-))[:k]` or b) briefly defining an $\text{argmax}_k(r(q,\cdot), D)$ that returns the function inputs $d\in D$ resulting in top k values of $r(q,d)$ and then using that.

>This analogy is further supported by the LLMs’ architecture. As illustrated in Figure 1, the generative modeling with LLMs can be interpreted as the matching process of a bi-encoder model ...

This analogy seems fair for the first generated token, but not for subsequent tokens, where the last layer representation being decoded depends on both the query and previously generated tokens of $y$.

In the equation on L131-136, is there an implicit assumption that the embedding and the embedding matrixes of the LLM are transposes of each other?

L154: since $r$ is already being used to denote re-ranker, it would be best to use something other than that for the "rule". For that matter $f_{\text{rerank}}$ could maybe just be $r$.

L255: it's not obvious to me as a reader how figure 2b supports this statement.

L301: "More details of how the temperature " typo.

**Other Strengths And Weaknesses:**

None beyond those discussed above.

**Questions For Authors:**

Have the ranking objectives (contrastive, list MLE, lambda rank) considered in this paper, been used for the direct purpose of training LLM based re-rankers (I saw RankT5 in the related work, but it doesn't cover all of the ranking objectives in this paper and the underlying models are somewhat different)? If so, is their a correlation between the following?

- the performance observed using a ranking objective for LLM-based reranker training
- the performance observed using a ranking objective for alignment tuning

**Relation To Broader Scientific Literature:**

The contributions are highly relevant to language model alignment (potentially also to language model tuning more broadly, i.e. for purposes other than being pleasant to chat with). It's possible they also have impact in applications of language models for information retrieval (by showing LLM alignment research has overlap with LLM ranking research).

**Theoretical Claims:**

No, I did not closely read the appendix. However, having read the DPO paper and the first few sections of this one, I readily believe the results linking ranking objective functions to their alignment-objective analogues.

---

> ### Author Rebuttal · Authors · 2025-03-31
>
> We sincerely thank the reviewer for the constructive feedback, which has strengthened our work. Please find our point-by-point responses below:
>
> - **Fig 2: Single dataset?** Thank you for the suggestion. We plan to incorporate this in the revised manuscript (e.g., using NQ for both 2(a) and 2(b)). However, note that Figure 2 is intended to highlight the analogy between retriever and LLM scaling behaviors, not to compare them directly, so using different datasets seems acceptable. We adopt NQ and GSM8K since NQ is a standard dataset for retrieval; GSM8K captures LLM reasoning capabilities.
>
> - **Fig 2: Metrics are monotonic.** We agree that Recall@N and Pass@N obviously increase with N. Our goal was to visualize similar trends in scaling behavior rather than make a strong claim. We will tone down the language in the revised manuscript.
>
> - **Fig 2: "Suboptimal" word choice.** We agree and will revise to replace "suboptimal."
>
> - **Table 2: DPO baseline with strong RM.** Thank you. We have now included iterative DPO with a strong reward model:
>
> | Model | ApEval 2 (LC WR) | ApEval 2 (WR) | MixEval | MixEval-Hard |
> |--|--|--|--|--|
> | **Mistral-Base (7B)** |
> | Iterative DPO | 32.85 | 35.2 | 0.6825 | 0.3835 |
> | LarPO (Contrastive) | 41.5 | 42.9 | 0.718 | 0.417 |
> | LarPO (LambdaRank) | 35.8 | 34.1 | 0.717 | 0.431 |
> | LarPO (ListMLE) | 36.6 | 37.8 | 0.73 | 0.423 |
> | **Mistral-Instruct (7B)** |
> | Iterative DPO | 38.95 | 47.89 | 0.698 | 0.3965 |
> | LarPO (Contrastive) | 43 | 53.8 | 0.718 | 0.425 |
> | LarPO (LambdaRank) | 41.9 | 48.1 | 0.74 | 0.44 |
> | LarPO (ListMLE) | 39.6 | 48.1 | 0.717 | 0.397 |
>
> LarPO consistently outperforms iterative DPO, validating its effectiveness with the strong reward model as well.
>
> - **Sec 6.1 vs. Table 2.** Section 6.1 isolates the effect of objectives under a fixed setup (e.g., hard negatives, candidate list construction) [1]. Table 2 reflects the joint effects of all factors. We’ll clarify this distinction in the revision.
>
> - **Fig 4a: Why GSM/Mathstral/iterative DPO?** We use GSM8K and Mathstral because such math problems have ground-truth labels, enabling clear negative type classification. Iterative DPO ensures a fixed candidate list to isolate the impact of negative difficulty.
>
> - **Fig 4b: Temperature vs. win rate.** We adjust the temperature to induce response hardness but agree other latent factors may play a role. We'll revise the explanation to focus on the importance of choosing an appropriate temperature rather than strong claims (“lower temperatures generate harder negatives”).
>
> - **Eq. 1–3: Missing document in set notation.** Thank you - we’ll update the notation accordingly with your suggestion (b).
>
> - **Fig 1: Analogy only partial.** Agreed. We’ll revise to clarify this is a similarity, not an identity. For later tokens, the prompt + decoded tokens act like a dynamic query in bi-encoder retrieval—but they differ fundamentally.
>
> - **L131–136: Embedding assumption.** Yes, we assume vocab embeddings and LLM hidden states share the same dimension, as in most LLMs. We’ll clarify this in the revision.
>
> - **r used for both re-ranker and rule.** Thank you - we’ll change the "rule" notation from r() to c() for clarity.
>
> - **L255: Fig 2b does not support the statement.** Thank you for the comment. The intended intuition is that lower temperatures yield more similar generated responses, increasing overlap between positive and negative samples. This effectively makes the negatives harder. We will revise the explanation to clarify this point in the revision.
>
> - **Correlation: reranker vs. alignment.** Great point. We believe a correlation may exist, but detailed exploration is beyond this paper’s scope and is an exciting direction for future work.
>
>
> [1] RLHF Workflow: From Reward Modeling to Online RLHF. TMLR 2024.

---

### Official Review · Reviewer_g1d5 · 2025-03-16

**Overall Recommendation:** 3

**Summary:**

The paper established a connection between LLM alignment and IR, particularly the retriever-reranker framework. With such a connection, the paper applied multiple techniques used in IR for LLM alignment, specifically, (1) IR objectives; (2) use of hard negatives (from a reasonably good model); (3) candidate list construction. Under these techniques, the paper proposed LARPO, which aims to iteratively optimize an LLM in a direct preference optimization way, so that it can learn to rank its generation responses according to a given reward model.  The paper was backed with extensive experiments against different baselines.

**Claims And Evidence:**

yes

**Essential References Not Discussed:**

No

**Experimental Designs Or Analyses:**

Yes, no issues observed.

**Methods And Evaluation Criteria:**

yes

**Other Comments Or Suggestions:**

Algorithm 1 line 13, $\mathcal D_s$ should be $\mathcal D_i$

**Other Strengths And Weaknesses:**

The paper is well motivated and has a great presentation to connect IR and LLM alignment, which is novel. Such a connection was not only motivated conceptually, but also inspired new algorithms for LLM equipped with existing techniques from IR.

However, as many techniques are applied for the optimization, it would be very interesting to see some ablation studies, say:
1. Without hard negatives, how's the performance?
2. Without inclusiveness/memorization, how's the performance?
3. How does the number of generated responses affect the performance?
4. Compared with traditional methods, how would the introduction of the candidate list affect the training total time?

**Questions For Authors:**

See Other Strengths And Weaknesses

**Relation To Broader Scientific Literature:**

The paper discussed the connection between IR and LLM, which is novel.

**Theoretical Claims:**

n/a

---

> ### Author Rebuttal · Authors · 2025-03-31
>
> We appreciate your insightful feedback, which has significantly strengthened our manuscript. We address each comment below.
>
> - **Without hard negatives, how's the performance?** Thank you for the question. As shown in Section 6.2 and Figure 4(a), removing hard negatives (i.e., using only easy/easiest ones) leads to significantly worse performance. This highlights the importance of hard negatives in enhancing LLM capabilities.
>
> - **Without inclusiveness/memorization, how's the performance?** Thank you for the question. We have conducted a detailed study of the inclusiveness and memorization in Figure 4(c) and Table 4. From Figure 4(c), an increase in the candidate list size can contribute to improved performance (inclusiveness). From Table 4, incorporating previous iteration responses can further enhance the LLM’s capability (memorization).
>
> - **How does the number of generated responses affect the performance?** The study of the number of generated responses can be found in Figure 4(c) and also shown below. We find that increasing the number of responses improves the win rate, indicating that our method can utilize and benefit from more diverse candidates:
>
> | Length | LC Winrate (%) | Winrate (%) |
> |--|--|--|
> | 2 | 49.75 | 55.07 |
> | 4 | 50.02 | 61.72 |
> | 6 | 52.56 | 63.59 |
> | 8 | 55.21 | 64.88 |
> | 10 | 55.52 | 64.42 |
>
> - **How does the candidate list affect training time?** We evaluate training time per iteration of LarPO (contrastive variant) with varying candidate list sizes:
>
> | Length | time (min) | Winrate (%) |
> |--|--|--|
> | 2 | 25 | 55.07 |
> | 4 | 45 | 61.72 |
> | 6 | 65 | 63.59 |
>
> We observe that larger candidate lists incur linearly higher costs but yield substantial performance gains, allowing users to balance efficiency and effectiveness trade-off.
>
> - **Algorithm 1 line 13.** Thank you for pointing this out. The operation merges all responses from previous iterations. We will revise accordingly.

---

### Decision · Program_Chairs · 2025-05-01

**Decision:**

Accept (poster)

**Comment:**

The paper makes an interesting connection between LM alignment and the retrieval-reranker IR framework. This allows the authors to propose a new method they call LARPO. It opens a range of new tricks for alignment including using hard negatives and candidate lists.

The paper is very well written, novel and connects two bodies of literature in a creative way that will enable more cross-proliferation and transfer of techniques. Experimentally the authors show good results on AlpacaEval2 and MixEval and compare to numerous prior techniques like DPO, SimPO and several others.
The authors also did a solid job in responding to reviewer questions by providing additional experiments to address comments and concerns. Overall a solid contribution.